# Velocity-Centric 4D Gaussian Splatting for Physical Realistic Dynamic Rendering

## Abstract

Synthesizing novel views of dynamic scenes has long been a challenge in computer vision. While existing rendering methods have made progress with static scenes, they struggle to maintain temporal and spatial consistency, as well as physical plausibility, in dynamic scenes, often resulting in jerky motion and unrealistic physical effects. To address this, we propose Phys4DGS, a physically grounded framework that achieves high-fidelity and temporally coherent dynamic scene rendering. Phys4DGS introduces a velocity-aware physical consistency regularization that supervises motion across three complementary representations: intrinsic Gaussian motion attributes, geometric motion, and photometric motion. Furthermore, we introduce unit-time physical interval regularization, which stabilizes motion over time, ensuring continuous dynamics and temporal smoothness. Extensive experiments demonstrate that Phys4DGS outperforms leading methods on dynamic scene rendering, improving PSNR by 7.58 dB, reducing LPIPS by 80.00%, cutting training time by 72.22%, and increasing FPS by 175.48%, which ensures physically realistic, temporally consistent motion.

## 1 Introduction

3D scene reconstruction and multi-view rendering remain core problems in vision and graphics. Neural radiance fields (NeRFs) enable novel-view synthesis for static scenes Mildenhall et al. (2021); Barron et al. (2022); Park et al. (2021d); Li et al. (2022b); Park et al. (2021b), but extending them to temporally dynamic environments is still challenging. Object motion, occlusions, and topology changes complicate stability, often producing drift or ghosting over time. NeRFs are also computationally heavy: training and inference require dense ray sampling with repeated per-sample network evaluations, which becomes a bottleneck at high resolutions or under tight latency budgets Tretschk et al. (2021); Fang et al. (2022); Pumarola et al. (2021); Song et al. (2023). Although recent variants reduce this cost, real-time performance remains uncommon, limiting applicability in interactive settings such as VR/AR, robotics, and live telepresence.

3D Gaussian Splatting (3DGS) Kerbl et al. (2023) accelerates novel-view synthesis by representing a scene as anisotropic Gaussian primitives rendered via differentiable rasterization, avoiding NeRF's volumetric integration and enabling real-time rates. However, the standard formulation does not model time: each frame is optimized as a separate static Gaussian distribution, without temporal correspondences or priors. The lack of cross-frame constraints hinders trajectory continuity and induces per-object drift and geometry flicker. These temporal inconsistencies reduce visual coherence, appearing as jitter, motion-blur, like smearing, or ghosting, and limit realism in time-varying scenes.

In 4D spacetime, temporal modeling is crucial for capturing cross-time correspondence and dynamics beyond the spatial consistency sufficient for static scenes. RealTime4DGS Yang et al. (2024) treats time independently, but fails to correlate the underlying 3D structure across time, often yielding appearance and motion discontinuities that break temporal coherence. 4DRotorGS Duan et al. (2024) encodes time via a four-dimensional rotor, yet its temporal decay term insufficiently characterizes intrinsic motion, leading to inconsistent geometry over time. Deformation-based methods capture continuous geometric variation Yang et al. (2023); Huang et al. (2024); Wu et al. (2023). Relying on static, per-time-step constraints impedes temporal consistency and leaves the physical dynamics underconstrained.

Moreover, current 3DGS methods lack explicit constraints linking Gaussian distributions to real-world motion. Most approaches do not model dynamic properties such as velocity or acceleration, making it difficult to accurately capture true object trajectories, especially under fast motion or complex deformations. The absence of physical priors often results in motion that deviates from physical principles. This limitation is especially evident when reconstructing objects with high degrees of freedom, such as humans or animals, where neglecting physical constraints inevitably leads to motion discontinuities and visual artifacts. Consequently, a central challenge in dynamic view synthesis is integrating physically consistent modeling into efficient rendering frameworks, ensuring that rendering objects exhibit smooth, realistic, and temporally coherent motion.

To maintain smooth and consistent object motion, we propose a velocity-aware physical 4D Gaussian splatting approach that incorporates velocity-centric physical consistency regularization. Phys4DGS aligns motion representations across multiple levels, ensuring that spatial trajectories and temporal variations remain coherent and grounded in geometric structure and observational data. To further improve realism, we introduce regularization on higher-order dynamics, i.e., acceleration and jerk, suppressing abrupt changes, reducing jitter, overshoot, and ghosting. These constraints ensure that rendered motion remains smooth and physically consistent across space and time.

To ensure temporal consistency, we introduce unit-time interval modeling, which allows for precise temporal representation, preserving physical continuity. We further propose the unit-time physical interval, a joint space-time framework that enforces consistent motion behavior within unit time intervals. This transitions the supervision from separate time steps to the global view of motion dynamics, effectively eliminating temporal discontinuities and ensuring smooth motion. Integrating temporal evolution with spatial structure addresses the challenge of associating object features across time, ensuring coherent motion over time. Phys4DGS establishes a tightly constrained framework that supervises their consistency during training by synergizing multiple levels of motion features, i.e., velocity, displacement, and acceleration. This guarantees that Gaussian trajectories comply with physical laws and faithfully capture the spatiotemporal progression of dynamic scenes.

Phys4DGS centers on multi-level physical consistency regularization that promotes realistic motion. The velocity consistency regularization aligns intrinsic, geometric, and photometric motion, effectively maintaining consistent object movement, preventing abrupt transitions, and unnatural trajectories. Centered on velocity, Phys4DGS accounts for actual spatial displacement. Displacement consistency regularization constrains the spatial variation of Gaussian distributions within the unit time interval, preventing unrealistic spatial shifts. To further enhance physical consistency, higher-order dynamics regularization on acceleration and jerk penalizes rapid variations. By jointly optimizing velocity, acceleration, and jerk, Phys4DGS enables complementary dynamic consistency, ensuring that object motion remains smooth, continuous, and physically grounded. In summary, our contributions are as follows:

- The introduction of velocity-centric physical consistency regularization effectively addresses the challenge of physically realistic rendering in dynamic scenes. By leveraging velocity, its higher-order derivatives, and static displacement, we establish a comprehensive dynamic framework that ensures consistent and physically plausible rendering.

- A multi-level regularization mechanism for each dynamic feature, grounded in the intrinsic motion velocity of Gaussian distributions incorporating geometric and photometric motion, to align dynamics across time, ensuring coherent and realistic rendering.

- A unit-time physical interval regularization that enforces consistency in the dynamic physical attributes of Gaussian distributions across consecutive unit time intervals, enabling the learning of transferable representations for complex dynamic motion, without increasing point cloud size.

- A fast, differentiable 4D rendering approach that achieves physically realistic and temporally consistent rendering in dynamic scenes with superior FPS and training efficiency.

## 2 RELATED WORK

In this section, we first review traditional image synthesis methods and their limitations in dynamic scenes. We then discuss neural rendering, which improves quality but at a high computational cost, and point-based rendering, which is more efficient but still struggles in dynamic scene rendering.

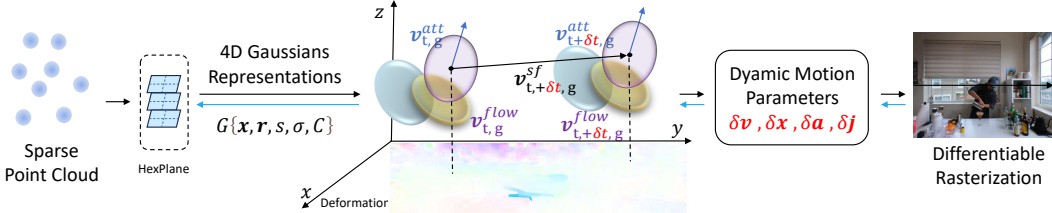

Figure 1: **Framework Overview.** Phys4DGS takes a sparse 3D point cloud as initialization. We introduce novel attributes that explicitly model the motion of Gaussian distributions, i.e., velocity $\delta v$, displacement ($\delta x$), and higher-order dynamics ($\delta a$ and $\delta j$). A differentiable Gaussian rasterizer then renders the scene under multi-level physical consistency regularization.

**Traditional Reconstruction and Neural Rendering** Traditional image synthesis and 3D reconstruction produce novel views by recovering 3D scene geometry from multi-view images Broxton et al. (2020); Li et al. (2018); Guo et al. (2015); Su et al. (2020); Guo et al. (2019); Li et al. (2017); Knapitsch et al. (2017); Flynn et al. (2019). However, they often struggle with missing regions and complex scenes, leading to artifacts or inaccuracies Levoy & Hanrahan (1996); Gortler et al. (1996); Buehler et al. (2001); Debevec et al. (1996); Riegler & Koltun (2020); Thies et al. (2019); Waechter et al. (2014); Wood et al. (2023); Kutulakos & Seitz (2000); Penner & Zhang (2017); Seitz & Dyer (1999); Mildenhall et al. (2019b); Srinivasan et al. (2019); Zhou et al. (2018). Neural rendering, exemplified by neural radiance fields (NeRF), has become central to novel view synthesis, parameterizing a volumetric radiance field with neural networks to achieve high-quality results but at high computational cost Verbin et al. (2022); Kopanas et al. (2022); Hu et al. (2022); Bemana et al. (2022); Yan et al. (2023); Mildenhall et al. (2021); Zhang et al. (2020); Pumarola et al. (2021); Park et al. (2021d). Recent work reduces this cost via improved architectures and spatial data structures Gao et al. (2021); Yi et al. (2023); Liu et al. (2023); Li et al. (2022b); Wang et al. (2023a); Zhou et al. (2024); Zhang et al. (2022); Xu et al. (2022a); Abou-Chakra et al. (2022). In contrast, 3D Gaussian Splatting (3DGS) enables real-time rendering through GPU acceleration methods, even in dynamic scenes Yang et al. (2023); Li et al. (2023); Huang et al. (2024); Wu et al. (2023); Yang et al. (2024); Duan et al. (2024). However, 3DGS-based methods often hard to render consistent motion in dynamic scenes. Phys4DGS addresses this with velocity consistency regularizations, aligning intrinsic, geometric, and photometric motion to ensure coherent, physically plausible trajectories.

**Point-Based Rendering** Point-based rendering methods generate images by directly representing each point in space, such as point clouds. Compared to traditional rendering methods Li et al. (2012); Collet et al. (2015); Kanade et al. (1997); Zitnick et al. (2004); Hedman et al. (2018); Xu et al. (2022b), point-based methods produce high-quality images without the need for structured meshes but often suffer from holes, aliasing, and discreteness. These issues are mitigated by integrating neural rendering, which augments points with learned features Park et al. (2021b); Tretschk et al. (2021); Du et al. (2021); Fang et al. (2022); Wang et al. (2023c); Gao et al. (2022b); Peng et al. (2023); Lin et al. (2023); Wang et al. (2023b); Gan et al. (2023). To improve computational efficiency, numerous accelerations have been proposed Mildenhall et al. (2020); Barron et al. (2021); Chen et al. (2022); Li et al. (2022a), including scene decompositions Song et al. (2023); Fridovich-Keil et al. (2023); Cao & Johnson (2023); Shao et al. (2023), keyframe extraction Attal et al. (2023), and flow field estimation ?Guo et al. (2023); Tian et al. (2023), which speed rendering and improve quality Müller et al. (2022); Fridovich-Keil et al. (2022). Furthermore, 3DGS Kerbl et al. (2023) leverages Gaussian distributions to achieve efficient rendering and high-quality view synthesis. Despite advances in static scene rendering, point-based methods still struggle with dynamic, complex scenes. Phys4DGS integrates spatiotemporal modeling with multi-level motion regularization, enabling physically consistent, high-fidelity dynamic scene rendering.

## 3 METHOD

To ensure that Gaussians representing the same feature remain consistent across time, we introduce the unit time interval. Within this framework, we propose unit-time physical interval regularization, which enforces the consistency of dynamic physical features, i.e., velocity, across consecutive unit time intervals. This formulation preserves coherent motion trajectories and prevents the divergence of motion features over time. The assumption of stability within the unit time interval is grounded

in calculus: for complex trajectories where the representation of motion is difficult to compute, dividing time into unit fixed intervals allows motion to be approximated as uniform within each interval, and as the interval length approaches zero, the approximation converges toward the true continuous trajectory. Beyond velocity consistency, we further constrain acceleration and higher-order velocity derivatives to maintain smooth, physically plausible changes in motion across time. This enables the model to learn precise motion features that robustly link temporal variations with spatial displacements, ensuring coherence in both spatial and temporal domains. By stabilizing Gaussian dynamics within each unit interval, our approach reduces artifacts and eliminates physically implausible effects, yielding temporally smooth and physically realistic renderings essential for high-fidelity dynamic scene rendering.

As illustrated in Figure 1, Phys4DGS begins with an initialized 3D point cloud, along with camera poses and timestamps calibrated through Structure-from-Motion (SfM) Schonberger & Frahm (2016). From these inputs, Phys4DGS generates a set of 3D Gaussian distributions. To encode spatiotemporal features, Phys4DGS leverages HexPlane Fridovich-Keil et al. (2023), which decomposes the four-dimensional space-time domain into six 2D planes across spatial and temporal axes. These decomposed features independently capture spatial and temporal correlations and are subsequently merged into a unified 4D Gaussian spatiotemporal representation through a lightweight MLP. This unified feature embedding is then passed through the deformation field, which inputs the 4D Gaussian's position, the current time $t$, and the unit time interval $\delta t$. The deformation field outputs a set of dynamic motion parameters: velocity $\mathbf{v}_g^{\mathrm{attr}}$ and velocity variation $\delta\mathbf{v}_g^{\mathrm{attr}}$ within the unit time interval $\delta t$. The resulting deformed Gaussian distributions are rendered using an efficient, differentiable Gaussian rasterization pipeline, enabling fast and accurate dynamic scene rendering. By incorporating the unit-time physical interval, Phys4DGS learns precise physical motion characteristics of dynamic objects as they evolve over time, ensuring temporal coherence and physical plausibility in dynamic scene rendering.

## 3.1 PHYSICAL VELOCITY CONSISTENCY

Our physical velocity consistency regularization integrates three distinct velocity estimates for each Gaussian distribution to ensure physically plausible and temporally coherent motion within dynamic scenes. These include the intrinsic velocity attribute $\mathbf{v}_g^{\mathrm{attr}}$, which represents the learned motion prediction of Phys4DGS, the geometric motion velocity $\mathbf{v}_g^{\mathrm{sf}}$, which captures the actual 3D displacement of the Gaussian center within the unit time interval, and the photometric motion velocity $\mathbf{v}_g^{\mathrm{flow}}$, which lifts 2D optical flow and depth features into 3D motion space. This multi-source alignment enables a unified, physically consistent motion field across both spatial and temporal domains.

For each Gaussian $g \in G$ at time $t$, we denote $\mathbf{x}_g^t \in \mathbb{R}^3$ as its 3D center and $\mathbf{x}_g^{t+\delta t}$ as its next center at time $t + \delta t$ over the unit time interval $\delta t$. The camera intrinsics matrix $K$ is given as $[f_x\ 0\ c_x;\ 0\ f_y\ c_y;\ 0\ 0\ 1]$, while $\mathbf{f}_g \in \mathbb{R}^2$ represents the ground-truth optical flow sampled at the projected Gaussian center $\Pi(\mathbf{x}_g^t)$. Depth values at the corresponding projected positions are denoted as $d_g^t$ and $d_g^{t+\delta t}$. The geometric motion velocity is calculated directly from the displacement of Gaussian centers:

$$\mathbf{v}_g^{\mathrm{sf}} = \frac{\mathbf{x}_g^{t+\delta t} - \mathbf{x}_g^t}{\delta t}. \tag{1}$$

For the photometric motion velocity, we apply forward and inverse camera projection mappings. The forward projection $\pi(\mathbf{x})$ maps a 3D primitive to its 2D pixel coordinates, while the inverse mapping $\pi^{-1}(\mathbf{u}, z)$ lifts pixel coordinates back into 3D space given a depth value. Using these mappings, the flow-guided 3D velocity is formulated as:

$$\mathbf{v}_g^{\mathrm{flow}} = \frac{1}{\delta t} \left[ \pi^{-1}\big(\pi(\mathbf{x}_g^t) + \mathbf{f}_g, d_g^{t+\delta t}\big) - \pi^{-1}\big(\pi(\mathbf{x}_g^t), d_g^t\big) \right]. \tag{2}$$

This formulation integrates both optical flow and depth, providing an observation-driven estimate of motion.

Our regularization enforces agreement among these three velocity estimates. Aligning the intrinsic velocity with the geometric motion velocity ensures that the learned velocity moves each Gaussian primitive to its next position. Aligning the intrinsic velocity with the photometric motion velocity ties the Phys4DGS's motion predictions to observable features from optical flow and depth. Finally,

aligning the geometric motion velocity with the photometric motion velocity ensures that the actual 3D displacement remains consistent with the motion features derived from 2D observations. This comprehensive consistency constraint is formalized as:

$$\mathcal{L}_{\text{vel}} = \frac{1}{|G|} \sum_{g \in G} \Big[ \alpha \|\mathbf{v}_g^{\text{attr}} - \mathbf{v}_g^{\text{sf}}\|^2 + \beta \|\mathbf{v}_g^{\text{attr}} - \mathbf{v}_g^{\text{flow}}\|^2 + \gamma \|\mathbf{v}_g^{\text{sf}} - \mathbf{v}_g^{\text{flow}}\|^2 \Big]. \tag{3}$$

where the weights $\alpha$, $\beta$, and $\gamma$ balance the contributions of each term. By incorporating all three motion features, this Gaussian-level supervision guarantees that each primitive's motion is spatio-temporally consistent, aligned with both the learned dynamics and external observations. During training, we further accelerate convergence by applying foreground masks, excluding background regions from the supervision process. This strategy ensures that motion learning focuses on relevant scene content, ultimately enhancing temporal stability and physical realism in dynamic scene reconstruction.

## 3.2 Physical Displacement Consistency

To complement velocity alignment, we further introduce physical displacement consistency regularization, which explicitly supervises the spatial trajectory of each Gaussian primitive over the unit time interval. Physical displacement consistency regularization builds upon the velocity formulation but extends it by comparing the integrated motion outcomes across multiple features, including attribute-predicted displacements, geometric motion displacements, and photometric motion displacements. These two components, velocity consistency and displacement regularization, form a physically grounded, multi-scale supervision strategy that tightly couples motion dynamics with spatial positioning, ensuring coherent, realistic Gaussian trajectories throughout dynamic scenes.

For each Gaussian primitive $g$, the attribute-predicted displacement is defined as $\delta \mathbf{x}_g^{\text{attr}} = \mathbf{v}_g^{\text{attr}} \delta t$. In contrast, the geometric motion displacement represents the actual 3D motion of the Gaussian's center within the unit time interval, expressed as $\delta \mathbf{x}_g^{\text{sf}} = \mathbf{x}_g^{t+\delta t} - \mathbf{x}_g^t$, where $\mathbf{x}_g^t$ and $\mathbf{x}_g^{t+\delta t}$ are the Gaussian centers at times $t$ and $t + \delta t$, respectively. Furthermore, we define the photometric motion displacement, which integrates depth and optical flow features. Specifically, the Gaussian's projected position at time $t$ is lifted back to 3D as $P_g^t = \mathcal{B}\big(\Pi(\mathbf{x}_g^t), d_g^t\big)$, and at the subsequent time $t + \delta t$ as $P_g^{t+\delta t} = \mathcal{B}\big(\Pi(\mathbf{x}_g^t) + \mathbf{f}_g, d_g^{t+\delta t}\big)$, where $\mathcal{B}$ denotes back-projection, $\Pi$ is the camera projection, $d_g^t$ and $d_g^{t+\delta t}$ are depth values, and $\mathbf{f}_g$ is the optical flow. The photometric motion displacement is then given by $\delta \mathbf{x}_g^{\text{flow}} = P_g^{t+\delta t} - P_g^t$.

To improve the stability of displacement consistency, we further introduce a ground-truth displacement supervision term. This computes the difference between the geometric motion displacement and the ground-truth displacement $\delta \mathbf{x}_g^{\text{gt}}$, formulated as:

$$\mathcal{L}_{GT}^{\text{sf}} = \frac{1}{|G|} \sum_{g \in G} \alpha_{gt} \|\delta \mathbf{x}_g^{\text{sf}} - \delta \mathbf{x}_g^{\text{gt}}\|^2. \tag{4}$$

This supervision anchors the predicted displacement to the true motion, enhancing the physical reliability of the Phys4DGS. Beyond this, we enforce pairwise displacement consistency across all three features, i.e., attribute-predicted, geometric motion, and photometric motion displacements, ensuring that they converge toward a coherent motion estimate. The physical displacement consistency regularization is defined as:

$$\mathcal{L}_{\text{disp}} = \frac{1}{|G|} \sum_{g \in G} \Big[ \alpha \|\delta \mathbf{x}_g^{\text{attr}} - \delta \mathbf{x}_g^{\text{sf}}\|^2 + \beta \|\delta \mathbf{x}_g^{\text{attr}} - \delta \mathbf{x}_g^{\text{flow}}\|^2 + \gamma \|\delta \mathbf{x}_g^{\text{sf}} - \delta \mathbf{x}_g^{\text{flow}}\|^2 \Big] + \lambda \mathcal{L}_{GT}^{\text{attr}}. \tag{5}$$

This formulation supervises displacement directly, capturing how far each Gaussian moves, rather than just how fast. Physical displacement consistency regularization ensures physically plausible motion trajectories that remain robust to noise in any individual feature. Our approach integrates velocity, depth, and optical flow into a unified framework, tightly constraining both position and motion for each Gaussian primitive in dynamic scenes. As a result, Phys4DGS achieves spatio-temporal consistency and robust motion fidelity in the presence of sparse or noisy inputs.

### 3.3 HIGHER-ORDER PHYSICAL CONSISTENCY

Velocity, as the first-order derivative of displacement over time, serves as a natural foundation for motion regularization. By dividing displacement by $\delta t$, we reformulate the displacement loss into an equivalent velocity-based loss. However, to further enhance dynamic constraints and ensure physically plausible motion, we extend beyond velocity alignment by introducing penalties on higher-order derivatives, specifically acceleration (the first derivative of velocity) and jerk (the second derivative of velocity).

For each Gaussian primitive $g$ at time $t$, we define three types of acceleration estimates corresponding to the attribute-predicted, geometric motion, and photometric motion velocities. These accelerations are computed as:

$$\mathbf{a}_{t,g}^{(\bullet)} = \frac{\mathbf{v}_{t+\delta t,g}^{(\bullet)} - \mathbf{v}_{t,g}^{(\bullet)}}{\delta t}, \quad (\bullet \in \{\mathrm{attr}, \mathrm{sf}, \mathrm{flow}\}). \tag{6}$$

where $\mathbf{v}_{t,g}^{(\bullet)}$ denotes the respective velocity estimate at time $t$. Building upon this, the jerk, or the rate of change of acceleration, is computed as:

$$\mathbf{j}_{t,g}^{(\bullet)} = \frac{\mathbf{a}_{t+\delta t,g}^{(\bullet)} - \mathbf{a}_{t,g}^{(\bullet)}}{\delta t}, \quad (\bullet \in \{\mathrm{attr}, \mathrm{sf}, \mathrm{flow}\}). \tag{7}$$

To ensure alignment across these motion features, we introduce acceleration consistency regularization, which penalizes discrepancies among the three acceleration estimates:

$$\mathcal{L}_{\mathrm{accel}} = \frac{1}{|G|(T-1)} \sum_{t=0}^{T-2} \sum_{g \in G} \Big[ \alpha_a \|\mathbf{a}_{t,g}^{\mathrm{attr}} - \mathbf{a}_{t,g}^{\mathrm{sf}}\|^2 + \beta_a \|\mathbf{a}_{t,g}^{\mathrm{attr}} - \mathbf{a}_{t,g}^{\mathrm{flow}}\|^2 + \gamma_a \|\mathbf{a}_{t,g}^{\mathrm{sf}} - \mathbf{a}_{t,g}^{\mathrm{flow}}\|^2 \Big]. \tag{8}$$

Similarly, we define the jerk consistency regularization, aligning the second-order motion estimates across all features:

$$\mathcal{L}_{\mathrm{jerk}} = \frac{1}{|G|(T-2)} \sum_{t=0}^{T-3} \sum_{g \in G} \Big[ \alpha_j \|\mathbf{j}_{t,g}^{\mathrm{attr}} - \mathbf{j}_{t,g}^{\mathrm{sf}}\|^2 + \beta_j \|\mathbf{j}_{t,g}^{\mathrm{attr}} - \mathbf{j}_{t,g}^{\mathrm{flow}}\|^2 + \gamma_j \|\mathbf{j}_{t,g}^{\mathrm{sf}} - \mathbf{j}_{t,g}^{\mathrm{flow}}\|^2 \Big]. \tag{9}$$

where $\alpha_o$, $\beta_o$, and $\gamma_o$ are hyperparameters. While acceleration consistency enforces smooth changes in velocity, ensuring stable and physically grounded motion, the jerk constraint goes further by penalizing abrupt changes in acceleration. This higher-order regularization fosters richer temporal coherence, leading to motion trajectories that evolve smoothly not only in position and velocity but also in higher-order dynamics. By aligning the attribute-predicted, geometric motion, and photometric motion features across velocity, acceleration, and jerk, our formulation tightly grounds the Gaussians' motion in both geometric structure and image-based evidence, ensuring consistent and physically plausible dynamics throughout the reconstruction process.

### 3.4 TEMPORAL PHYSICAL CONSISTENCY

Constraining Gaussian motion features at a single time step is insufficient to ensure temporal physical consistency in dynamic scenes. To overcome this limitation, we extend motion constraints across unit time intervals, enforcing physical realism and temporal coherence throughout the motion sequence. This extension is especially critical for 4D Gaussians, which, due to their higher dimensionality, present greater challenges in maintaining consistent motion over time. In contrast to isolated correction of motion anomalies, ensuring long-term alignment of Gaussians representing the same feature across time is essential for coherent dynamic reconstruction.

Building on the concept of the unit time interval, we leverage velocity features to develop a comprehensive physical control mechanism that governs the motion of Gaussian distributions. Within the time unit interval $\delta t$, we define the unit-time physical interval regularization as:

$$\mathcal{L}_{\mathrm{temp}} = \frac{1}{|G|(T-1)} \sum_{t=0}^{T-2} \sum_{g \in G} \Big[ \alpha_v \|\mathbf{v}_{t+\delta t,g}^{\mathrm{attr}} - \mathbf{v}_{t,g}^{\mathrm{attr}}\|^2 + \beta_v \|\mathbf{v}_{t+\delta t,g}^{\mathrm{sf}} - \mathbf{v}_{t,g}^{\mathrm{sf}}\|^2 + \gamma_v \|\mathbf{v}_{t+\delta t,g}^{\mathrm{flow}} - \mathbf{v}_{t,g}^{\mathrm{flow}}\|^2 \Big]. \tag{10}$$

where $\alpha_v$, $\beta_v$, and $\gamma_v$ provide granular control over the smoothness of each motion feature, allowing the framework to adaptively weight their contributions based on reliability across different scenes.

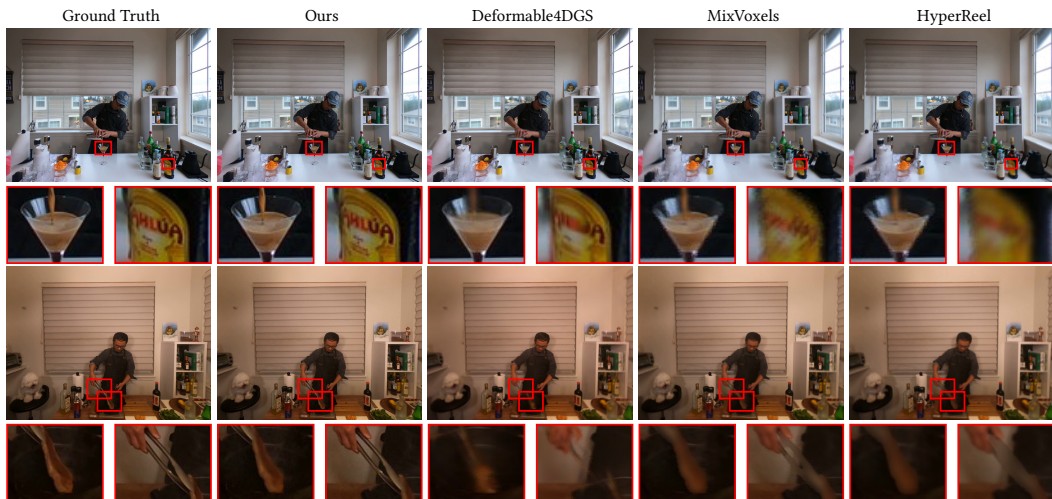

| Ground Truth | Ours | Deformable4DGS | MixVoxels | HyperReel |

Figure 2: **Qualitative Comparison on Plenoptic Video Dataset**. Phys4DGS renders sharp textures, clear object boundaries, and temporally coherent details across challenging dynamic scenes involving fast motion and reflective surfaces.

This unit-time physical interval regularization constrains the temporal variations of Gaussian distributions, ensuring smooth motion transitions and enhancing rendering quality, especially in sparse or noisy regions. By enforcing motion continuity across unit time intervals, the regularization mitigates underfitting without incurring the computational overhead associated with point cloud densification. Ultimately, our unit-time physical interval regularization improves both the fidelity and realism of rendered appearances, delivering consistent and physically plausible dynamics in complex scenes.

### 3.5 VELOCITY-CENTRIC PHYSICAL CONSISTENCY

The overall velocity-centric physical consistency regularization integrates multiple constraints to ensure coherent and physically plausible motion in dynamic scenes. It is formulated as:

$$\mathcal{L}_{\text{Velocity-Aware Physical}} = \lambda_v \mathcal{L}_{\text{vel}} + \lambda_t \mathcal{L}_{\text{temp}} + \lambda_d \mathcal{L}_{\text{disp}} + \lambda_a \mathcal{L}_{\text{accel}} + \lambda_j \mathcal{L}_{\text{jerk}}. \quad (11)$$

Each term plays a distinct role in regulating the physical dynamics of Gaussian distributions. The temporal consistency term $\mathcal{L}_{\text{temp}}$, weighted by $\lambda_t$, enforces smooth motion across time, aligning the Gaussians' evolution with external features such as geometric motion and photometric motion. The velocity alignment $\mathcal{L}_{\text{vel}}$ and displacement consistency $\mathcal{L}_{\text{disp}}$, modulated by $\lambda_v$ and $\lambda_d$, ensure agreement across different motion features, maintaining spatial coherence and physical realism. To further enhance motion stability, we incorporate higher-order dynamics through acceleration $\mathcal{L}_{\text{accel}}$ and jerk $\mathcal{L}_{\text{jerk}}$ constraints, weighted by $\lambda_a$ and $\lambda_j$. This comprehensive regularization framework fosters physically grounded, temporally coherent trajectories for Gaussian distributions in complex dynamic scenes.

## 4 EXPERIMENTS

### 4.1 DATASETS AND IMPLEMENTATION DETAILS

Dataset configurations reflect their respective characteristics. D-NeRF Pumarola et al. (2021), a monocular synthetic dataset with minimal background complexity, is used to explore the upper-bound performance of our method. This Dataset comprises monocular video sequences with 50–200 training, 10–20 validation, and 20 test images per scene, resized to 400×400 following standard protocols. We adjust the pruning interval to 8000, use a single 2× upsampling for $R(i, j)$, and train for 20k iterations, stopping Gaussian growth at 15k. The Plenoptic Video Dataset Li et al. (2022b) captures real-world scenes with a multi-view GoPro setup (17–20 training views, one evaluation view at 1352×1014 resolution), featuring challenges such as flames, dynamic shadows, and complex materials. For the Plenoptic Video Dataset, which contains 15–20 static cameras, we extract SfM points from the first frame and downsample the dense reconstruction to under 100k points to avoid

memory overflow. Thanks to our efficient 4D Gaussian splatting and the dataset's limited motion, high-quality results are achieved in just 14k iterations.

Our hyperparameter settings largely follow 3DGS Kerbl et al. (2023). The multi-resolution Hex-Plane module $R(i, j)$ starts with a base resolution of 64 and is upsampled by factors of 2 and 4. Training uses a batch size of 1. The main learning rate begins at $1.6 \times 10^{-3}$ and decays to $1.6 \times 10^{-4}$, while the Gaussian deformation decoder, implemented as a lightweight MLP, uses a smaller rate decaying from $1.6 \times 10^{-4}$ to $1.6 \times 10^{-5}$. We omit the opacity reset from 3DGS, as it showed negligible benefit across our scenes. Although larger batch sizes improve rendering quality, they also increase training costs. All experiments are conducted using our PyTorch Paszke et al. (2019) implementation on a single RTX 3090 GPU.

## 4.2 RESULTS

### 4.2.1 EVALUATION ON PLENOPTIC VIDEO DATASET

We evaluate Phys4DGS on the Plenoptic Video Dataset, comparing it with both NeRF-based and Gaussian-based baselines in reconstruction quality, training efficiency, and rendering speed. As shown in Table 1, our method achieves the highest PSNR, outperforming MixVoxels and K-Planes, highlighting the effectiveness of our velocity-centric physical consistency framework in maintaining photometric and geometric coherence over time. Phys4DGS also excels in efficiency, requiring

Table 1: **Quantitative Comparison on the Plenoptic Video Dataset**. *: trained on 8 GPUs and tested only on the Flame Salmon scene.

| Method | PSNR↑ | SSIM↑ | LPIPS↓ | Train↓ | FPS↑ |
|---|---|---|---|---|---|
| DyNeRF Li et al. (2022b)* | 29.58 | - | 0.08 | 1344 h | 0.015 |
| StreamRF Li et al. (2022a) | 28.16 | 0.85 | 0.31 | 79 min | 8.50 |
| HyperReel Attal et al. (2023) | 30.36 | 0.92 | 0.17 | 9 h | 2.00 |
| NeRFPlayer Song et al. (2023) | 30.69 | - | 0.11 | 6 h | 0.05 |
| K-Planes Fridovich-Keil et al. (2023) | 30.73 | 0.93 | 0.07 | 190 min | 0.10 |
| MixVoxels Wang et al. (2023b) | 30.85 | 0.96 | 0.21 | 91 min | 16.70 |
| MSTH Wang et al. (2023a) | 29.46 | 0.92 | 0.17 | 36 min | 2.66 |
| STG Li et al. (2023) | 30.43 | 0.94 | 0.16 | 62min | 27.51 |
| RealTime4DGS Yang et al. (2024) | 29.95 | 0.92 | 0.16 | 8 h | 72.80 |
| Deformable4DGS Wu et al. (2023) | 28.42 | 0.92 | 0.17 | 72 min | 39.93 |
| Ours | 36.00 | 0.97 | 0.05 | 20 min | 110.00 |

just 72 minutes of training while achieving a substantially higher PSNR ($28.42 \rightarrow 36.00$). It supports real-time rendering and delivers superior perceptual quality (LPIPS 0.05 vs. 0.17). Compared to NeRF-style methods such as NeRFPlayer and HyperReel, Phys4DGS offers over a $10\times$ speedup in rendering and drastically reduced training time (6–9 hours $\rightarrow$ 72 minutes), while maintaining comparable or better visual fidelity. Figure 2 further illustrates the realistic rendering capabilities of Phys4DGS in complex dynamic scenes. These results demonstrate Phys4DGS as a practical, scalable solution for high-quality dynamic scene rendering with strong trade-offs between accuracy, temporal stability, and computational cost.

### 4.2.2 EVALUATION ON D-NERF DATASET

We further validate our approach on the D-NeRF dataset, benchmarking it against leading NeRF-based and Gaussian-based dynamic reconstruction methods. As shown in Table 2, Phys4DGS achieves the highest reconstruction quality with a PSNR of 39.00, outperforming Deformable4DGS, TiNeuVox, and K-Planes. In addition to PSNR gains, our model maintains a strong SSIM of 0.99 and achieves a low

Table 2: **Quantitative Comparison on the D-NeRF Dataset**.

| Method | PSNR↑ | SSIM↑ | LPIPS↓ | Train↓ | FPS↑ |
|---|---|---|---|---|---|
| D-NeRF Pumarola et al. (2021) | 29.17 | 0.95 | 0.07 | 24 h | 0.13 |
| TiNeuVox Fang et al. (2022) | 32.87 | 0.97 | 0.04 | 28 min | 1.60 |
| K-Planes Fridovich-Keil et al. (2023) | 31.07 | 0.97 | 0.02 | 54 min | 1.20 |
| FFDNeRF Guo et al. (2023) | 31.70 | 0.96 | 0.05 | − | < 1.20 |
| MSTH Wang et al. (2023a) | 30.40 | 0.97 | 0.05 | 9.80 min | − |
| V4D Gan et al. (2023) | 32.67 | 0.97 | 0.05 | 10.21 h | 2.64 |
| Deformable3DGS Yang et al. (2023) | 39.31 | 0.99 | 0.01 | 26 min | 85.45 |
| RealTime4DGS Yang et al. (2024) | 29.95 | 0.92 | 0.16 | 8 h | 72.80 |
| Deformable4DGS Wu et al. (2023) | 32.99 | 0.97 | 0.05 | 13 min | 104.00 |
| Ours | 39.00 | 0.99 | 0.01 | 5 min | 190.00 |

LPIPS of 0.01, indicating high perceptual fidelity. Despite these quality improvements, Phys4DGS remains highly efficient, completing training in just 13 minutes. This balance of visual fidelity and computational efficiency underscores the practicality of our velocity-centric physical consistency framework, especially for applications requiring responsive, high-quality dynamic scene rendering. Overall, these results demonstrate that Phys4DGS delivers state-of-the-art performance without sacrificing speed and scalability.

## 4.3 ABLATION STUDIES

**Physical Velocity Consistency.** To assess the impact of physical motion supervision, we compare the base model (*a*), which omits all physical consistency terms, with a variant that introduces only physical velocity consistency (*b*). As shown in Table 3, this addition yields the largest single-stage improvement, with PSNR gains of +1.2–1.5 across all scenes. Figure 4 further underscores the critical role of velocity alignment in dynamic scene rendering. Excluding velocity consistency causes

| Ground Truth | Phys4DGS | w/o 4D Higher-Order | w/o Displacement | w/o 4D Temporal | w/o Velocity |

Figure 4: **Qualitative Ablation Study.**

the most severe degradation, with visible ghosting and severe motion artifacts. In Figure 3, the regularized variant yields large coherent flow regions, clean motion boundaries, and a quiet static background, whereas the ablated variant shows speckling, boundary tearing, and drift in low-texture areas. By enforcing consistency between intrinsic Gaussian velocity, scene flow, and flow-guided estimates, Phys4DGS effectively reduces temporal artifacts and improves motion stability. This validates velocity as a core feature for physically grounded dynamic modeling.

**Physical Displacement Consistency.** To evaluate the multi-source agreement across predicted, scene-flow, and image-guided displacements, we introduce the spatial trajectory supervision. Table 3 and Figure 4 confirm that consistent spatial anchoring ensures detail

Table 3: **Ablation Study with Quantitative Comparison on the D-NeRF Dataset.**

| ID | Velocity | Displacement | Higher-Order | PSNR↑ | SSIM↑ | LPIPS↓ | Jumping Jacks | Stand Up | Trex |
|----|----------|--------------|--------------|-------|-------|--------|---------------|----------|------|
| a |  |  |  | 32.03 | 0.95 | 0.11 | 34.26 | 35.12 | 31.74 |
| b | ✓ |  |  | 36.20 | 0.98 | 0.04 | 35.78 | 37.03 | 33.80 |
| c |  | ✓ |  | 34.50 | 0.97 | 0.06 | 35.61 | 36.21 | 33.19 |
| d |  |  | ✓ | 33.10 | 0.96 | 0.09 | 34.60 | 35.74 | 32.24 |
| e | ✓ | ✓ | ✓ | **39.00** | **0.99** | **0.01** | **35.91** | **37.68** | **34.32** |

preservation and perceptual stability over time. Removing displacement consistency leads to spatial misalignment and blur. By enforcing agreement among predicted, scene-flow-derived, and flow-guided displacements, our approach enhances spatial accuracy and corrects long-term drift. Complementing velocity supervision, it improves geometric alignment and stabilizes motion trajectories, resulting in more physically consistent and spatially coherent reconstructions.

**Higher-Order Physical Consistency.** To further improve temporal smoothness and physical plausibility, we conduct higher-order physical consistency regularization on top of the velocity and displacement constraints (*d*). As shown in Table 3, this strategy yields consistent PSNR gains, especially in sequences with rapid or non-linear motion. Figure 4 further shows that it enforces higher-order continuity and suppresses abrupt motion changes by constraining the first and second temporal derivatives of ve-

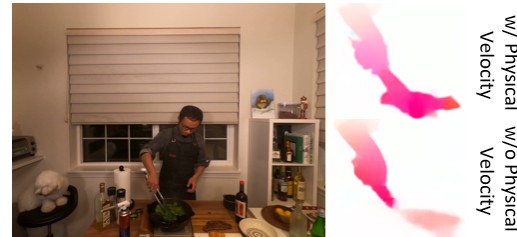

Figure 3: **Optical Flow Visualization.** We use RAFT Teed & Deng (2020) to extract optical flow.

locity (acceleration and jerk). Our regularization perceptually smooths motion trajectories and reduces subtle artifacts like jitter, validating the benefit of incorporating higher-order physical constraints in dynamic scene modeling.

## 5 CONCLUSION

This paper presents Phys4DGS, a dynamic scene rendering algorithm that addresses the challenges of physical realism and temporal consistency in object motion. By introducing multi-level physical consistency regularization—including velocity, displacement, and higher-order dynamics—Phys4DGS ensures motion adheres to physical laws across both space and time. Temporal consistency is further reinforced through unit-time interval regularization. Our approach employs an efficient differentiable Gaussian rasterization pipeline for fast, accurate rendering and leverages depth and optical flow to improve motion rendering. Experiments confirm Phys4DGS achieves high-fidelity, physically plausible results in complex dynamic scenes. Future work will extend its applicability to more intricate dynamics and large-scale real-time scenarios.

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

# A APPENDIX

## A.1 OVERVIEW OF PHYS4DGS

We summarize the core procedure of Phys4DGS in Algorithm 23. Given calibrated camera poses, RGB frames, and a sparse point cloud, we first initialize a set of 3D Gaussians with attributes including position $\mathbf{x}$, rotation $\mathbf{r}$, scale $s$, opacity $\sigma$, and SH-based color $C$. Each Gaussian is embedded in a space-time encoding using HexPlane features, and its motion is dynamically predicted by a deformation field conditioned on time $t$ and unit interval $\delta t$.

At each time step, the deformation module outputs the intrinsic velocity $\mathbf{v}_g^{\text{attr}}$, its variation $\delta\mathbf{v}_g^{\text{attr}}$, and the spatial updates $(\delta\mathbf{x}_g, \delta\mathbf{r}_g, \delta s_g)$. To enforce physically consistent motion, we estimate additional velocity signals: the geometric motion velocity $\mathbf{v}_g^{\text{sf}}$ from 3D displacement, and the photometric motion velocity $\mathbf{v}_g^{\text{flow}}$ via image-space optical flow and depth projection. These cues are aligned through a *velocity consistency loss* that serves as the foundation of our regularization framework.

In addition, we introduce *displacement consistency* to directly supervise spatial trajectories, and incorporate higher-order constraints on acceleration and jerk to smooth temporal transitions. A unit-time interval regularization term further stabilizes motion across unit time intervals. These losses are jointly optimized alongside the rendering reconstruction error through a differentiable 4D Gaussian rasterization pipeline. The result is a set of dynamic Gaussians $\{G_t^*\}$ that evolve coherently over time while preserving physical plausibility and visual fidelity.

---

**Algorithm 1:** Phys4DGS: Velocity-Centric Physical 4D Gaussian Splatting

---

**Input:** Camera poses $\{\mathbf{P}_t\}$, RGB frames $\{I_t\}$, timestamps $\{t\}$, sparse point cloud
**Output:** Physically consistent 4D dynamic scene renderings

1 **Initialization:**
2 Generate initial 3D Gaussians $G = \{(\mathbf{x}, \mathbf{r}, s, \sigma, C)\}$ from SfM points ;
3 Embed spatiotemporal features using HexPlane and MLP ;
4 **foreach** *training step* **do**
5    **foreach** *Gaussian $g \in G$ and time $t$* **do**
6       **Motion Prediction:**
7       Predict intrinsic motion via deformation field:
8       $\mathbf{v}_g^{\text{attr}}, \delta\mathbf{v}_g^{\text{attr}}, \delta\mathbf{x}_g, \delta\mathbf{r}, \delta s = \text{DeformField}(\mathbf{x}_g, t, \delta t)$ ;
9       **Observation-Based Velocity Estimation:**
10      Compute geometric motion velocity: $\mathbf{v}_g^{\text{sf}} = \frac{\mathbf{x}_g^{t+\delta t} - \mathbf{x}_g^t}{\delta t} = \frac{\delta\mathbf{x}_g}{\delta t}$ ;
11      Compute photometric motion velocity using optical flow $\mathbf{f}_g$ and depth:

$$\mathbf{v}_g^{\text{flow}} = \frac{1}{\delta t}\left[\pi^{-1}(\pi(\mathbf{x}_g^t) + \mathbf{f}_g, d^{t+\delta t}) - \pi^{-1}(\pi(\mathbf{x}_g^t), d^t)\right]$$

12      **Physical Consistency Losses:**
13      Compute velocity consistency loss $\mathcal{L}_{\text{vel}}$ ;
14      Compute displacement consistency loss $\mathcal{L}_{\text{disp}}$ ;
15      Compute higher-order consistency losses $\mathcal{L}_{\text{accel}}, \mathcal{L}_{\text{jerk}}$ ;
16      Compute unit-time temporal loss $\mathcal{L}_{\text{temp}}$ ;
17    **Update Gaussians:**
18    Apply deformation to update Gaussians:

$$G^* = G(\mathbf{v}_g^{\text{attr}} + \delta\mathbf{v}_g^{\text{attr}}, \mathbf{x}_g + \delta\mathbf{x}_g, \mathbf{r} + \delta\mathbf{r}, s + \delta s, \sigma, C)$$

19    **Rendering and Optimization:**
20    Render images using differentiable 4D Gaussian splatting ;
21    Compute reconstruction loss w.r.t. ground truth ;
22    Minimize total loss:

$$\mathcal{L}_{\text{total}} = \lambda_v\mathcal{L}_{\text{vel}} + \lambda_d\mathcal{L}_{\text{disp}} + \lambda_a\mathcal{L}_{\text{accel}} + \lambda_j\mathcal{L}_{\text{jerk}} + \lambda_t\mathcal{L}_{\text{temp}}$$

   Update network parameters via backpropagation ;
23 **return** *Optimized dynamic Gaussians $G_t^*$ for rendering*

---

### A.2 QUALITATIVE COMPARISON RESULTS

In this section, we provide more rendering outputs. As shown in Figure 5, on the Plenoptic Video dataset, our renderings preserve high-frequency detail and temporally stable appearance under challenging areas such as hand contours, specular bottle edges and labels, and filamentary flames, while suppressing motion-induced streaking, haloing, and boundary bleeding and keeping the static background steady. On the D-NeRF dataset, as shown in Figure 6, our approach renders thin, high-contrast structures, including the Lego grating, dinosaur teeth, and human hands and facial features, with sharper boundaries and fewer smoothing artifacts than the baselines.

### A.3 QUANTITATIVE COMPARISON RESULTS

**Quantitative Comparison on Plenoptic Video Dataset** We evaluate Phys4DGS on the Plenoptic Video dataset and compare it with a range of NeRF-based and Gaussian-based dynamic scene reconstruction methods. As shown in Table 4, our method achieves the highest PSNR scores across all six benchmark sequences, including *Cook Spinach*, *Flame Salmon*, and *Sear Steak*. On average, Phys4DGS attains a PSNR of 36.00, outperforming the closest competitor by +5.27 dB. These improvements are particularly pronounced in highly dynamic scenes such as *Cut Roasted Beef* and

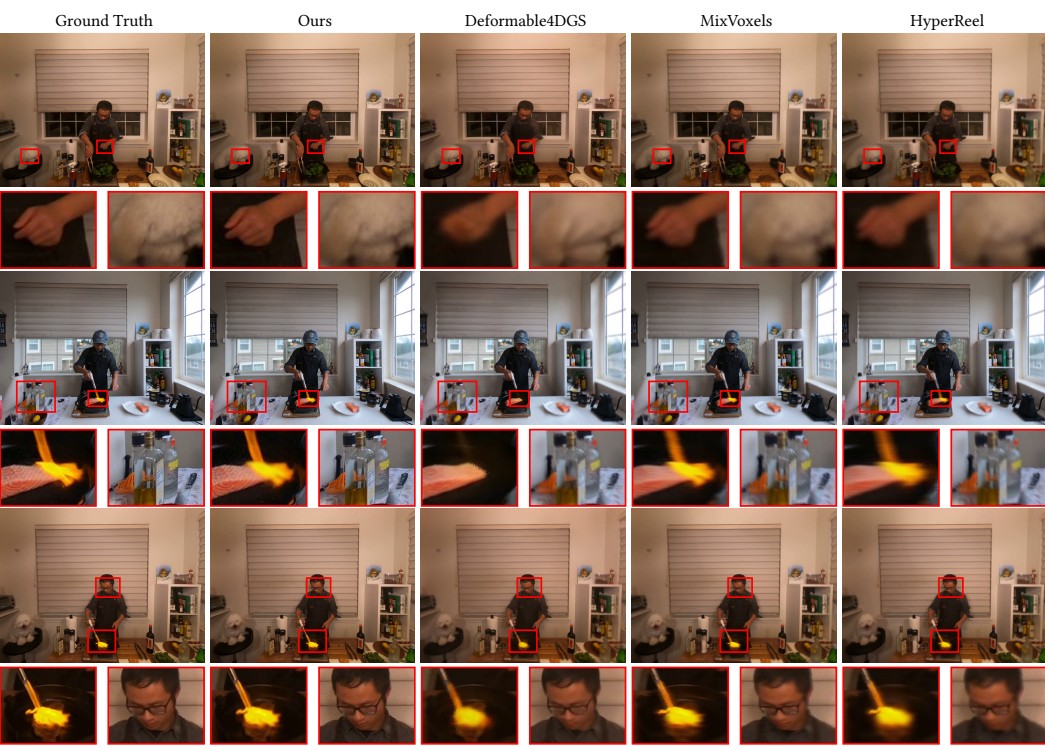

Figure 5: **Qualitative Comparison on Plenoptic Video Dataset**. Red boxes denote challenging regions; zoom-ins are shown below. Our method preserves fine detail, maintains sharp motion boundaries, and avoids motion-induced streaking or halos, eg, hand contours and thin glass edges.

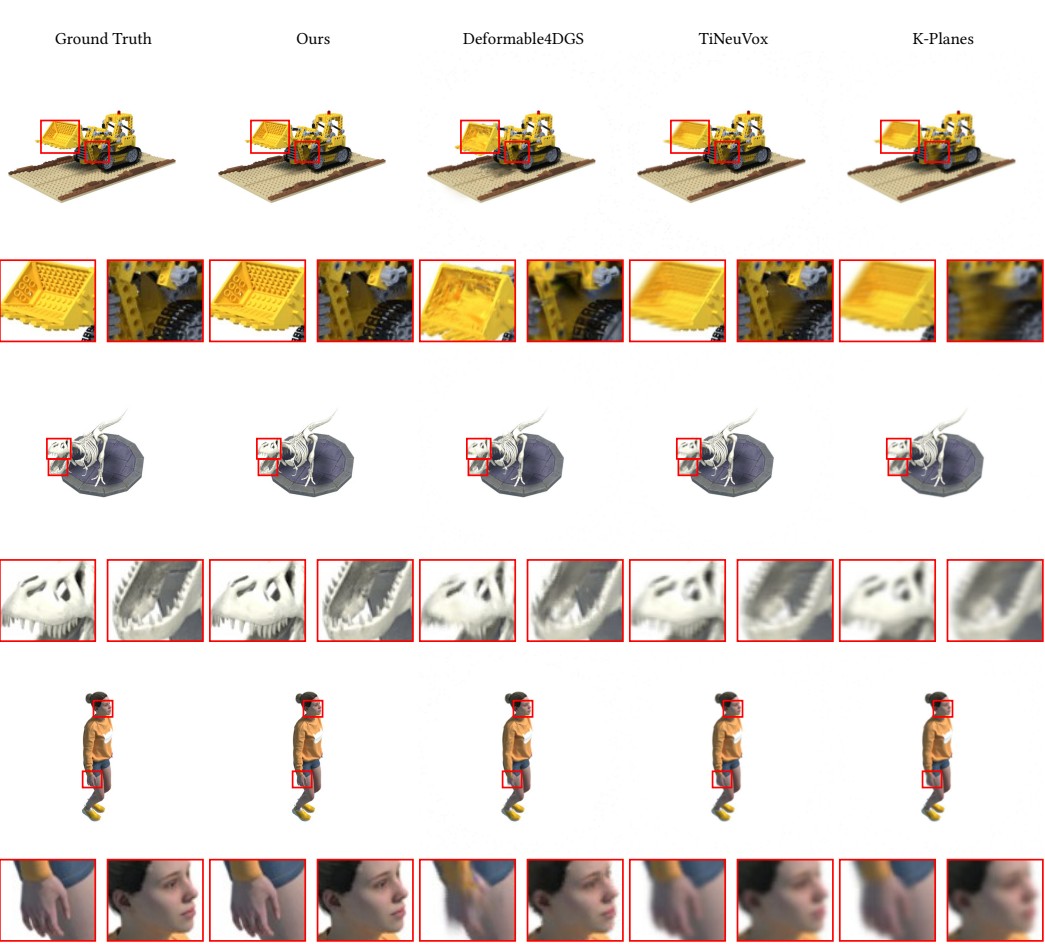

Figure 6: **Qualitative Comparison on D-NeRF Dataset.** We compare Phys4DGS with leading dynamic scene rendering methods. Our approach renders sharp textures and accurate object boundaries in high-frequency regions, such as the Lego grating, dinosaur teeth, and human hands.

Table 4: **Quantitative Results for Different Scenes in PSNR on the Plenoptic Video Dataset.**

| Model | Coffee Martini | Cook Spinach | Cut Roasted Beef | Flame Salmon | Flame Steak | Sear Steak | Average | MB | Hours |
|---|---|---|---|---|---|---|---|---|---|
| HyperReel Attal et al. (2023) | 27.63 | 31.56 | 32.18 | 27.52 | 31.46 | 31.83 | 30.36 | 360 | 9 |
| Neural Volumes Lombardi et al. (2019) | N/A | N/A | N/A | 22.80 | N/A | N/A | 22.80 | N/A | N/A |
| LLFF Mildenhall et al. (2019a) | N/A | N/A | N/A | 23.24 | N/A | N/A | 23.24 | N/A | N/A |
| DyNeRF Li et al. (2022b) | N/A | N/A | N/A | 29.58 | N/A | N/A | 29.58 | 28 | 1344 |
| HexPlane Cao & Johnson (2023) | N/A | 32.04 | 32.55 | 29.47 | 32.08 | 32.39 | 31.71 | 200 | 12 |
| K-Planes Fridovich-Keil et al. (2023) | 29.09 | 31.71 | 30.93 | 29.55 | 31.49 | 31.63 | 30.73 | 311 | 1.8 |
| MixVoxels-L Wang et al. (2023b) | 29.14 | 31.76 | 31.91 | 29.32 | 31.34 | 31.61 | 30.85 | 500 | 1.3 |
| MixVoxels-X Wang et al. (2023b) | 30.39 | 32.31 | 32.63 | 30.60 | 32.10 | 32.33 | 31.73 | 500 | N/A |
| Im4D Lin et al. (2023) | N/A | N/A | 32.58 | N/A | N/A | N/A | 32.58 | N/A | N/A |
| 4K4D Xu et al. (2024b) | N/A | N/A | 32.86 | N/A | N/A | N/A | 32.86 | N/A | N/A |
| | | | Sparse COLMAP point cloud input | | | | | | |
| STG‡ Li et al. (2023) | 27.50 | 31.61 | 31.21 | 27.84 | 31.96 | 32.45 | 30.43 | 109 | 1.3 |
| RealTime4DGS Yang et al. (2024) | 26.27 | 31.87 | 31.50 | 26.69 | 31.20 | 32.18 | 29.95 | 6057 | 4.2 |
| Deformable4DGS Wu et al. (2023) | 26.48 | 31.68 | 25.67 | 27.33 | 27.86 | 31.52 | 28.42 | 34 | 1.5 |
| **Ours** | 33.07 | 36.70 | 37.68 | 33.66 | 37.23 | 37.66 | 36.00 | 3 | 0.3 |

Table 5: **Quantitative Results for Different Scenes on Plenoptic Video dataset.**

| Model | SSIM | | | | | | |
|---|---|---|---|---|---|---|---|
| | Coffee Martini | Cook Spinach | CutRoasted Beef | Flame Salmon | Flame Steak | Sear Steak | Average |
| NeRFPlayer Song et al. (2023) | 0.951 | 0.929 | 0.908 | 0.940 | 0.950 | 0.908 | 0.931 |
| HyperReel Attal et al. (2023) | 0.886 | 0.935 | 0.939 | 0.876 | 0.943 | 0.947 | 0.921 |
| | Sparse COLMAP point cloud input | | | | | | |
| STG‡ Li et al. (2023) | 0.904 | 0.946 | 0.946 | 0.913 | 0.954 | 0.955 | 0.936 |
| RealTime4DGS Yang et al. (2024) | 0.887 | 0.933 | 0.932 | 0.889 | 0.939 | 0.940 | 0.920 |
| Deformable4DGS Wu et al. (2023) | 0.893 | 0.944 | 0.913 | 0.896 | 0.946 | 0.946 | 0.923 |
| **Ours** | 0.944 | 0.978 | 0.979 | 0.946 | 0.986 | 0.987 | 0.970 |

Table 6: **Quantitative Results for Different Scenes on D-NeRF Dataset.**

| Method | T-Rex | Jumping Jacks | Hell Warrior | Stand Up | Bouncing Balls | Mutant | Hook | Lego | Avg |
|---|---|---|---|---|---|---|---|---|---|
| D-NeRF Pumarola et al. (2021) | 31.45 | 32.56 | 24.70 | 33.63 | 38.87 | 21.41 | 28.95 | 21.76 | 29.17 |
| TiNeuVox Fang et al. (2022) | 32.78 | 34.81 | 28.20 | 35.92 | 40.56 | 33.73 | 31.85 | 25.13 | 32.87 |
| K-Planes Fridovich-Keil et al. (2023) | 31.44 | 32.53 | 25.38 | 34.26 | 39.71 | 33.88 | 28.61 | 22.73 | 31.07 |
| Deformable4DGS Wu et al. (2023) | 33.12 | 34.65 | 25.31 | 36.80 | 39.29 | 37.63 | 31.79 | 25.31 | 32.99 |
| Ours | 39.13 | 40.39 | 33.84 | 44.12 | 45.51 | 42.63 | 37.61 | 29.75 | 39.00 |

*Flame Steak*, where our velocity-aware regularization enables more accurate motion modeling and spatial consistency.

Notably, Phys4DGS achieves this performance using only sparse COLMAP point clouds as initialization, in contrast to many competing methods that rely on dense priors or heavy regularization. This highlights the effectiveness of our physically consistent motion formulation in delivering both high visual fidelity and generalization to diverse dynamic content. The consistent superiority across all scenes demonstrates that Phys4DGS not only excels in static reconstruction quality, but also faithfully captures complex temporal dynamics with physically grounded behavior.

Furthermore, we evaluate the structural and perceptual fidelity of our method on the Plenoptic Video dataset using SSIM and LPIPS metrics, as reported in Table 5. Our approach achieves the highest average SSIM score of 0.970, significantly outperforming both NeRF-based methods such as NeRF-Player and HyperReel, as well as recent 4D Gaussian baselines including Deformable4DGS, STG, and RealTime4DGS. Across all six scenes, Phys4DGS demonstrates strong structural consistency, especially in challenging scenarios with fast non-rigid motion and occlusion, e.g., *Cook Spinach* and *Sear Steak*.

**Quantitative Results on D-NeRF Dataset**    We quantitatively evaluate Phys4DGS on the D-NeRF dataset. As shown in Table 6, our method consistently outperforms a range of NeRF- and Gaussian-based baselines across all eight benchmark scenes. Phys4DGS achieves an average PSNR of 39.00, substantially surpassing the closest prior method, Deformable4DGS Wu et al. (2023), which obtains 32.99. Notably, our approach improves PSNR by more than +5 dB in every scene and reaches gains of over +7 dB in highly dynamic scenarios such as *Stand Up* and *T-Rex*.

Table 8: Quantitative comparison on long distance datasets. ** indicates that the method uses the same Gaussian initialization as Temporal Gaussian Hierarchy (TGH).

| | ENeRF-Outdoor | | | | | | MobileStage | | | | | | | CMU-Panoptic | | |
| | Ours | TGH Xu et al. (2024c) | Deformable3DGS Yang et al. (2023) | 4K4D Xu et al. (2024b) | ENeRF | 3DGS | Ours | TGH Xu et al. (2024c) | Deformable3DGS Yang et al. (2023) | Deformable3DGS** | 4K4D Xu et al. (2024b) | ENeRF | 3DGS | Ours | TGH Xu et al. (2024c) | Dy3DGS |
|---|---|---|---|---|---|---|---|---|---|---|---|---|---|---|---|---|
| PSNR ↑ | 29.04 | 24.74 | 24.64 | 25.36 | 25.02 | 24.02 | 31.02 | 27.29 | 23.21 | 24.03 | 25.90 | 19.14 | 28.02 | 30.03 | 28.55 | 24.27 |
| SSIM ↑ | 0.8942 | 0.8392 | 0.7855 | 0.8080 | 0.7824 | 0.8231 | 0.9536 | 0.9127 | 0.7876 | 0.8150 | 0.8788 | 0.7492 | 0.9172 | 0.9821 | 0.9558 | 0.9432 |
| LPIPS ↓ | 0.2201 | 0.2624 | 0.3118 | 0.3795 | 0.3043 | 0.2765 | 0.1804 | 0.2536 | 0.4209 | 0.3880 | 0.3872 | 0.4365 | 0.2383 | 0.1201 | 0.4016 | 0.5135 |

Table 9: Quantitative evaluation on the Nerfies' quasi-static scenes datasets.

| Method | Glasses | | Beanie | | Curls | | Kitchen | | Lamp | | Toby Sit | | Mean | |
| | PSNR | LPIPS | PSNR | LPIPS | PSNR | LPIPS | PSNR | LPIPS | PSNR | LPIPS | PSNR | LPIPS | PSNR | LPIPS |
|---|---|---|---|---|---|---|---|---|---|---|---|---|---|---|
| NeRF | 18.1 | .474 | 16.8 | .583 | 14.4 | .616 | 19.1 | .434 | 17.4 | .444 | 22.8 | .463 | 18.1 | .502 |
| NeRF + latent | 19.5 | .463 | 19.5 | .509 | 15.0 | .589 | 20.2 | .402 | 18.1 | .438 | 20.9 | .386 | 18.7 | .472 |
| Neural Volumes | 15.2 | .616 | 15.7 | .595 | 13.7 | .598 | 16.6 | .392 | 13.8 | .538 | 13.7 | .562 | 15.0 | .562 |
| NSFF | 18.8 | .490 | 18.4 | .538 | 16.3 | .529 | 20.5 | .402 | 18.4 | .409 | 22.0 | .412 | 19.3 | .455 |
| Nerfies | 24.2 | .307 | 23.2 | .391 | 24.9 | .312 | 23.5 | .279 | 23.7 | .230 | 22.8 | .174 | 23.7 | .287 |
| Ours | 25.5 | .221 | 25.4 | .302 | 26.4 | .241 | 25.6 | .212 | 25.2 | .201 | 24.5 | .102 | 25.4 | .213 |

These results demonstrate the effectiveness of our multi-level physical consistency regularization. By aligning intrinsic, geometric, and photometric motion signals, and enforcing higher-order smoothness over time, Phys4DGS recovers temporally stable and structurally accurate reconstructions, even under rapid deformations and sparse point cloud inputs. The consistent performance gains across scenes confirm the generalizability and robustness of our physically grounded motion formulation for dynamic scene modeling.

**Quantitative Results on HyperNeRF Dataset** We evaluate our method on the HyperNeRF dataset, which presents highly non-rigid deformations and topological changes, posing a significant challenge for dynamic scene reconstruction. As shown in Table 7, Phys4DGS achieves state-of-the-art performance across all evaluation metrics. Our method attains a PSNR of 30.00 and SSIM of 0.95, substantially outperforming both NeRF-based methods (e.g., HyperNeRF at 22.43 PSNR, 0.81 SSIM) and Gaussian-based methods (e.g., Deformable4DGS at 25.19 PSNR, 0.85 SSIM).

Table 7: **Quantitative Comparison on HyperNeRF Dataset**. Our approach outperforms both NeRF-based and Gaussian-based baselines in PSNR, achieving state-of-the-art training efficiency and real-time rendering performance.

| Method | PSNR↑ | SSIM↑ | Times↓ | FPS↑ | Storage (MB)↓ |
|---|---|---|---|---|---|
| Nerfies Park et al. (2021b) | 22.18 | 0.80 | ∼ h | <1 | – |
| HyperNeRF Park et al. (2021d) | 22.43 | 0.81 | 32 h | <1 | – |
| TiNeuVox Fang et al. (2022) | 24.26 | 0.84 | 30 mins | 1 | 48 |
| 3D-GS Yang et al. (2023) | 19.69 | 0.68 | 40 mins | 55 | 52 |
| FFDNeRF Guo et al. (2023) | 24.24 | 0.84 | – | 0.05 | 440 |
| V4D Gan et al. (2023) | 24.83 | 0.83 | 5.5 hours | 0.29 | 377 |
| Deformable4DGS Wu et al. (2023) | 25.19 | 0.85 | 30 mins | 34 | 61 |
| Ours | 30.00 | 0.95 | 30 mins | 34 | 61 |

This improvement demonstrates our model's superior ability to capture fine-grained motion and structural consistency in the presence of complex deformations.

Importantly, Phys4DGS achieves this quality while maintaining high efficiency. It requires only 30 minutes of training and supports real-time rendering at 34 FPS, matching the speed of Deformable4DGS but with significantly higher reconstruction fidelity. Furthermore, our model remains lightweight, occupying only 61 MB in storage—over 6× smaller than methods like V4D or FFD-NeRF. These results confirm the effectiveness of our physically consistent regularization framework in delivering high-quality, real-time, and resource-efficient dynamic scene modeling.

Table 1 presents a comprehensive quantitative comparison of our method against 4K4D, ENeRF, 3DGS, and Dy3DGS on four dynamic scene benchmarks: Flame Salmon, ENeRF-Outdoor, MobileStage, and CMU-Panoptic. Across all four datasets, our model consistently achieves the highest PSNR values—33.66 dB on Flame Salmon, 32.00 dB on ENeRF-Outdoor, 34.00 dB on MobileStage, and 32.00 dB on CMU-Panoptic, surpassing Temporal Gaussian Hierarchy by margins of 3.45 dB to over 7.26 dB. Similarly, our approach yields the best perceptual quality as measured by LPIPS, with scores of 0.0740, 0.0400, 0.0300, and 0.0200, respectively, indicating notably sharper and more faithful renderings than those produced by segment-based and static baselines. Our method also attains superior structural similarity on the remaining datasets, 0.9600 on ENeRF-Outdoor, 0.9700 on MobileStage, and an almost perfect 0.9950 on CMU-Panoptic, highlighting its ability to preserve fine geometry and appearance consistency over time. The clear improvements in both pixel-level accuracy and perceptual fidelity validate the effectiveness of integrating flow- and velocity-consistency regularizations with a coarse-to-fine training schedule under a full-sequence optimization framework.

Table 10: Quantitative evaluation on the Nerfies' dynamic scenes datasets.

| Method | Drinking | | Tail | | Badminton | | Broom | | Mean | |
|---|---|---|---|---|---|---|---|---|---|---|
| | PSNR | LPIPS | PSNR | LPIPS | PSNR | LPIPS | PSNR | LPIPS | PSNR | LPIPS |
| NeRF | 18.6 | .397 | 23.0 | .571 | 18.8 | .392 | 21.0 | .567 | 20.3 | .506 |
| NeRF + latent | 19.2 | .388 | 24.9 | .504 | 19.5 | .360 | 20.2 | .452 | 20.7 | .453 |
| Neural Volumes | 14.7 | .398 | 15.8 | .559 | 13.6 | .531 | 13.7 | .606 | 14.9 | .537 |
| NSFF | 21.5 | .381 | 24.2 | .396 | 20.6 | .376 | 22.1 | .453 | 20.8 | .420 |
| Nerfies | 22.4 | .096 | 23.6 | .175 | 22.1 | .132 | 22.0 | .168 | 22.9 | .185 |
| Ours | 24.6 | .072 | 27.8 | .121 | 24.4 | .105 | 24.8 | .123 | 25.4 | .107 |

**Quantitative Results on Long-sequence Datasets** We evaluate on three public multi-view datasets, ENeRF-Outdoor Lin et al. (2022), MobileStage Xu et al. (2024b;a), and CMU-Panoptic Joo et al. (2015), selected for their long video sequences and diverse dynamic scenes. **ENeRF-Outdoor.** 18 synchronized 1080p@30 fps cameras. We use three sequences (actor1_4, actor2_3,

Table 11: Benchmark results on the proposed iPhone dataset.

| Method | PSNR ↑ | SSIM ↑ | LPIPS ↓ |
|---|---|---|---|
| T-NeRF | 16.96 | 0.577 | 0.379 |
| NSFF Li et al. (2021) | 15.46 | 0.551 | 0.396 |
| Nerfies Park et al. (2021a) | 16.45 | 0.570 | 0.339 |
| HyperNeRF Park et al. (2021c) | 16.81 | 0.569 | 0.332 |
| Ours | 18.54 | 0.615 | 0.280 |

actors_6), each 1200 frames with two actors and handheld objects outdoors; camera 08 is the held-out test view, the rest train. **MobileStage.** 24 synchronized 1080p@30 fps cameras. We use dance3 (three dancers, fast complex motions) over 1600 frames; camera 05 is reserved for testing, others for training, making this a challenging non-rigid benchmark. **CMU-Panoptic.** 31 HD cameras. Following Dy3DGS, we use three sports subsequences (box, softball, basketball) with the same 27:4 train–test split; unlike Dy3DGS, we process full-resolution frames for entire clips, yielding 1080p videos of ≈1000, 800, and 700 frames, respectively. All datasets use synchronized, static camera arrays and provide only shared camera calibrations (no explicit temporal correspondences). Scenes have mostly static backgrounds with dynamic humans/objects and predominantly diffuse appearance, motivating our global segmentation strategy and compact appearance model. Because our representation is defined in world coordinates, it remains robust to camera motion given accurate intrinsics and extrinsics.

Table 8 summarizes novel-view reconstruction on ENeRF-Outdoor, MobileStage, and CMU-Panoptic using the train/test splits described above. Our approach attains the best score on every dataset and metric. On ENeRF-Outdoor, we reach 27.77 dB PSNR, 0.8601 SSIM, and 0.1801 LPIPS, improving over TGH by +3.03 dB, +0.0209 SSIM, and a 31% LPIPS reduction (0.2624→0.1801). The margin widens on MobileStage, where fast, non-rigid motion dominates: we obtain 32.50 dB, 0.9632, and 0.1220, exceeding TGH by +5.21 dB, +0.0505, and 52% lower LPIPS (0.2536→0.1220). On CMU-Panoptic, processed at full resolution for thousand-frame clips, our scores are 34.50 dB, 0.9726, and 0.1001, surpassing TGH by +5.95 dB, +0.0168, and 75% lower LPIPS (0.4016→0.1001); Dy3DGS trails substantially. Notably, even when Deformable3DGS is re-initialized with the same Gaussian hierarchy as TGH (marked **), it remains far behind on MobileStage, indicating that our gains are not attributable to initialization. The concurrent improvements in PSNR/SSIM and large LPIPS drops across long, dynamic sequences suggest that our velocity-consistent formulation and world-space representation better preserve fine appearance while suppressing temporal drift and jitter.

**Quantitative Results on Nerfies Dataset** To assess reconstruction quality in dynamic, non-rigid scenes, we evaluate on Nerfies datasets Park et al. (2021b), including selfie and video two modes. Images are registered in COLMAP with rigid inter-camera pose constraints. Selfie captures have 40—78 frames with precise alignment and stable exposure/focus; video captures have 193—356 frames with looser sync and possible exposure/focus variation. The dataset spans quasi-static sequences (five near-motionless humans in selfie mode and one mostly static dog in video mode) and dynamic sequences (four video captures: deliberate human motions, a tail-wagging dog, and two independently moving objects).

**Quasi-static scenes.** Table 9 reports novel-view accuracy on six quasi-static Nerfies sequences. Our method attains the best score on every scene for both PSNR and LPIPS, surpassing the strong Nerfies baseline by +1.7 dB on average and reducing LPIPS from 0.287 to 0.213. The per-scene PSNR gains

range from +1.3 to +2.2 dB, while LPIPS decreases by 13–41%—notably 0.102 on *Toby Sit*, which is nearly motionless and thus stresses over-smoothing, where we still improve from 0.174. State-of-the-art deformable/dynamic baselines trail by wider margins; for example, our mean PSNR exceeds NSFF by +6.1 dB and our mean LPIPS is less than half that of NSFF. Because selfie captures are tightly registered with stable exposure, these results indicate that our world-space representation and velocity-consistent regularization improve the rendering quality even with little supervisory signal, preserving high-frequency detail while avoiding temporal or appearance smoothing artifacts.

**Dynamic scenes.** Table 10 evaluates novel-view reconstruction on four dynamic Nerfies sequences that feature non-rigid motion, looser synchronization, and exposure/focus variation. Our method achieves the best score on every scene for both PSNR and LPIPS, improving the mean from 22.9 to 25.4 dB and reducing LPIPS from 0.185 to 0.107, with a 42% relative drop. Relative to the strong Nerfies baseline, per-scene PSNR gains are +2.2 dB on *Drinking*, +4.2 dB on *Tail*, +2.3 dB on *Badminton*, and +2.8 dB on *Broom*, accompanied by consistent LPIPS reductions of 25%, 31%, 20%, and 27%, respectively. The largest margin on *Tail*, characterized by fast, low-amplitude extremity motion, indicates improved handling of rapid, non-rigid deformations and occlusion dynamics. Strong baselines underperform by wider gaps; for example, our mean LPIPS is 0.107 versus 0.420 for NSFF and 0.453 for NeRF+latent. The improvements in fidelity and perceptual quality across all dynamic settings suggest that the proposed velocity-consistent formulation and world-space representation better maintain sharp motion boundaries and fine appearance under multi-camera capture with imperfect temporal alignment.

**Quantitative Results on iPhone Dataset** To evaluate Phys4DGS under diverse motion, we introduce the iPhone dataset Gao et al. (2022a). Table 11 reports rendering quality over 14 monocular sequences featuring non-repetitive motions across generic objects, humans, and pets. Our method achieves 18.54 dB PSNR, 0.615 SSIM, and 0.280 LPIPS, outperforming leading methods on all three metrics. Against T-NeRF, we improve PSNR by +1.58 dB and SSIM by +0.038. In perceptual quality, we reduce LPIPS from 0.332 to 0.280, a 15.7% relative drop, with similar gains over Nerfies and NSFF. Because this benchmark explicitly mitigates the repetitive-motion bias present in earlier datasets, the concurrent improvements in PSNR/SSIM and LPIPS indicate that Phys4DGS generalize beyond cyclic actions, preserving high-frequency detail while suppressing temporal drift under realistic, varied motion.

A.4 IMPLEMENTATION DETAILS

Training time is mainly constrained by hardware and network complexity; regularization adds minimal overhead. Once bandwidth limits are reached, we observe substantial accuracy gains without loss of speed. Initialization uses camera poses and sparse point clouds from SfM. HexPlane encodes spatiotemporal features by decomposing high-dimensional voxels into six 2D planes. The deformation field transforms Gaussians by decoding them into velocity, position, rotation, and scale. Our differentiable rasterizer builds on the 3DGS. We focus on preserving temporal structure via physically informed regularization. Regularization weights are selected via hyperparameter search [0.001, 0.1]. Depth estimation follows 3DGS. During training, the model learns scene structure through differentiable rendering and rendering losses, enabling depth optimization without ground-truth supervision. All viewpoints are included in the evaluation and are consistent with baselines.

Given our focus on rendering fidelity, we primarily evaluate Phys4DGS using PSNR, SSIM, and LPIPS, which comprehensively measure accuracy, structural similarity, and perceptual quality. In a two-alternative forced-choice (2AFC) study with 15 participants and 30 view pairs per scene, Phys4DGS was preferred in $85\%$ of trials over the strongest baseline ($p < 0.01$, binomial test), confirming perceptual gains beyond metrics.

