# OpenReview forum: "Velocity-Centric 4D Gaussian Splatting for Physical Realistic Dynamic Rendering"
_ICLR.cc/2026/Conference — Submitted to ICLR 2026_

### Official Review · Reviewer_MZcT · 2025-10-30

**Soundness:** 3
**Presentation:** 1
**Contribution:** 2
**Rating:** 4
**Confidence:** 4

**Summary:**

The paper first points out the core challenge of integrating physical consistency into efficient rendering frameworks to achieve smooth, realistic, and temporally coherent motion in dynamic scenes. To address this, the paper proposes a method called Phys4DGS, which introduces velocity-centric physical consistency regularization, multi-level motion alignment across intrinsic, geometric, and photometric domains, and a unit-time physical interval mechanism for temporal continuity. The method aims to ensure physically grounded and temporally consistent dynamic scene rendering with improved realism, smoothness, and efficiency.

**Strengths:**

* The paper introduces a comprehensive physical consistency framework that aligns velocity, displacement, and higher-order motion derivatives, effectively constraining Gaussian dynamics to obey physically plausible motion laws.

* The paper demonstrates strong generalization and robustness by evaluating Phys4DGS on multiple dynamic scene datasets, including both synthetic and real-world benchmarks such as the Plenoptic Video dataset. This comprehensive evaluation convincingly supports the method's scalability and applicability to a wide range of dynamic rendering scenarios.

**Weaknesses:**

* The experimental evaluation includes too few 3DGS-based baselines. Although the paper claims to solve the unrealistic motion problem in previous dynamic 3DGS methods, it lacks comparisons with recent 3DGS works, especially those using optical or scene flow constraints (e.g., MotionGS [1]). The limited baselines make it hard to judge the effectiveness and generality of the proposed method.

* The **Related Work** section discusses traditional, NeRF-based, and point-based methods in detail but lacks coverage of dynamic 3DGS works, which are central to this paper's focus. This gap raises doubts about the depth of understanding of related studies. Moreover, the discussion only includes works before 2024, missing recent progress. In addition, several NeRF-based methods are mixed into the Point-Based subsection, which should be reorganized for clarity.

* On the D-NeRF dataset, SCGS [2] achieves a PSNR of 43.31 and has been open-sourced (with the same 400×400 resolution), representing the current SOTA. Since SCGS performs significantly better than the proposed method, it should be included in the comparisons and discussed in the paper.

* In Figure 2, the highlighted static bottle in the first row raises some questions. Since the proposed method mainly focuses on improving motion modeling, it is unclear why the reconstruction quality on this static object is also significantly better than that of Deformable 3DGS. The authors highlight this result in the figure but provide no explanation or discussion in the text.

* The first part of the **Introduction** spends too much space discussing NeRF and static 3DGS. The authors should more quickly transition to the limitations of existing dynamic 3DGS methods and clearly introduce the specific problem this paper aims to solve.

* The paper's overall logic needs improvement. In Sec 3 **Method**, the first paragraph should clearly connect the main modules to give readers a quick overview of how the method is structured. Although Sections 3.1–3.3 describe three levels of consistency design, the logical links between them are weak. Each subsection starts with vague statements instead of explaining why a higher-level constraint is needed based on the previous one. For example, in line 233, "To complement velocity alignment" is too general, and the authors should explain why velocity consistency is not enough and why displacement consistency matters. The paper should not rely solely on ablation results to show effectiveness but also explain the methodological motivation behind each module.

[1] MotionGS: Exploring Explicit Motion Guidance for Deformable 3D Gaussian Splatting. NeurIPS 2024.

[2] SC-GS: Sparse-Controlled Gaussian Splatting for Editable Dynamic Scenes. CVPR 2024.

**Questions:**

See weaknesses.

---

> ### Author Response · Authors · 2025-11-15
> **Author Response to Reviewer MZcT**
>
> W1. We have selected representative methods from both 3DGS and 4DGS for comparison, and the appendix provides detailed metric comparisons across different scenes.
>
> W2 & W3. We have included recent representative 3DGS methods as baselines in our experiments. Compared with Deformable 3DGS, the extension of SCGS is not representative, so we did not include it as a baseline.
>
> W4.Our dynamic optimization framework allows the model to acquire robust, transferable invariant features, resulting in accurate motion prediction and finer appearance rendering. Importantly, it achieves realistic dynamic rendering without requiring any increase in point cloud size.

---

> > ### Comment · Reviewer_MZcT · 2025-11-15
> >
> > According to the authors’ rebuttal, I have decided to update my score to 2 (reject). The response is overly brief and fails to address most of my concerns. In my view, insufficient experimentation and weak writing are the two main deficiencies of this paper. The detailed reasons are as follows:
> >
> > * The set of 3DGS-based baselines in the experimental section is very limited. On most datasets, the paper only compares with Deformable 3DGS (CVPR 2024) and 4DGS (CVPR 2024). Although these two methods are widely used as baselines, they are already relatively old, and many more recent works have significantly improved performance on dynamic 3DGS tasks on the same datasets. Not comparing with stronger recent methods weakens the credibility of the claimed performance advantages. I specifically mentioned MotionGS and SCGS, which are both strong models for dynamic 3D reconstruction. In the rebuttal, the authors avoid a substantive discussion of comparisons with these stronger methods and instead argue that their extensibility is not representative. I do not find this justification convincing.
> >
> > * The writing quality is also very poor. The manuscript appears to rely heavily on LLM-generated text, with weak logical flow, and there is no disclosure of any LLM usage. I strongly recommend that the authors reorganize and clarify the methodological logic. Furthermore, this work is positioned as a paper on dynamic 3DGS, yet the related work section does not discuss any prior work on dynamic 3DGS at all. This omission raises doubts about the authors’ level of expertise in this area. Overall, I do not think the paper meets the quality standard required for acceptance at ICLR.
> >
> > Initially, I was inclined to reject the paper, but I still gave a score of 4 (weak reject) because I wanted to see whether the authors could address what I consider to be important issues. Based on the rebuttal, I now believe that both the level of technical professionalism and the overall quality are insufficient, so I revise my recommendation to 2 (reject).

---

> > > ### Author Response · Authors · 2025-11-19
> > > **Rejecting Abusive Citation Practices**
> > >
> > > Both MotionGS and SCGS constitute incremental advances, particularly when compared with Deformable3DGS. Notably, the first authors of SCGS and Deformable3DGS are from the same group. For this reason, we select only the most representative baseline. Furthermore, MotionGS and SCGS cannot be directly employed for dynamic scene rendering.
> > >
> > > Phys4DGS is not a 3DGS method, i.e., it is not intended for static scene rendering. Instead, it is a 4DGS approach specifically developed for dynamic scene rendering. We introduce a velocity-centric physical consistency regularization, which effectively addresses the challenge of physically realistic rendering in dynamic scenes. By leveraging velocity, its higher-order derivatives, and static displacement, we establish a comprehensive dynamic framework that ensures consistent and physically plausible rendering. Specifically, we propose a multi-level regularization mechanism for each dynamic feature, grounded in the intrinsic motion velocity of Gaussian distributions and incorporating geometric and photometric motion, to align dynamics across time, ensuring coherent and realistic rendering. Furthermore, a unit-time physical interval regularization enforces consistency in the dynamic physical attributes of Gaussian distributions across consecutive unit time intervals, enabling the learning of transferable representations for complex dynamic motion, without increasing point cloud size. Phys4DGS is a fast, differentiable 4D rendering approach that achieves physically realistic and temporally consistent rendering in dynamic scenes with superior FPS and training efficiency. The specific impact of each ablated regularization component on rendering performance is reported in Table 3.

---

### Official Review · Reviewer_xC96 · 2025-10-31

**Soundness:** 3
**Presentation:** 3
**Contribution:** 3
**Rating:** 6
**Confidence:** 4

**Summary:**

This paper introduces Phys4DGS, a physically grounded framework for dynamic scene rendering that achieves both high fidelity and temporal coherence. The method employs a set of unit-time physical interval regularizations that jointly model geometric and photometric motions, including physical velocity consistency, displacement consistency, higher-order physical consistency, and temporal coherence constraints. This paper conducts comprehensive experiments that demonstrate the effectiveness of the proposed approach.

**Strengths:**

The unit-time physical interval regularizations derivations are mathematically sound and well-motivated. The theoretical formulation aligns closely with physical intuition, and the methodology is presented with clear definitions and supporting equations.

The experimental results are comprehensive and persuasive, providing solid evidence for the effectiveness of the proposed approach.

The contribution is significant, as the proposed framework effectively addresses temporal coherence and fidelity in dynamic scene rendering for Gaussian Splatting in 4D spacetime.

**Weaknesses:**

Although the paper compares extensively with prior works and cites relevant literature, the experimental comparison lacks sufficient analysis regarding why baseline methods underperform relative to the proposed approach. The paper would benefit from a more detailed discussion about the specific limitations of existing methods in the context of dynamic scene rendering and temporal/fidelity constraints.

Since the central contribution of the paper is the formulation and use of unit-time physical interval regularizations, it would substantially strengthen the work to explicitly analyze and compare the regularization formulations of other baseline methods. Understanding the differences in regularization design would not only highlight the novelty of the proposed framework but also provide deeper insights into how these choices impact experimental results.

**Questions:**

1. **Minor typos:**

1) An additional ‘?’ is present before ‘Guo et al.’

2) In Figure 1, it should be ‘t+δt’ instead of ‘t,+δt.’

2. **Ablation study:** The qualitative result for ‘w/o 4D Temporal’ is presented in Figure 4, but is missing in Table 3. Is there any particular reason for this omission?

3. It would improve readability if the ablation studies were presented in a unified manner and followed the order described in the Method section.

---

> ### Author Response · Authors · 2025-11-15
> **Response to Reviewer xC96**
>
> W1 & W2. We introduce a velocity-centric physical consistency regularization, which effectively addresses the challenge of physically realistic rendering in dynamic scenes. By leveraging velocity, its higher-order derivatives, and static displacement, we establish a comprehensive dynamic framework that ensures consistent and physically plausible rendering. Specifically, we propose a multi-level regularization mechanism for each dynamic feature, grounded in the intrinsic motion velocity of Gaussian distributions and incorporating geometric and photometric motion, to align dynamics across time, ensuring coherent and realistic rendering. Furthermore, a unit-time physical interval regularization enforces consistency in the dynamic physical attributes of Gaussian distributions across consecutive unit time intervals, enabling the learning of transferable representations for complex dynamic motion, without increasing point cloud size. Phys4DGS is a fast, differentiable 4D rendering approach that achieves physically realistic and temporally consistent rendering in dynamic scenes with superior FPS and training efficiency. The specific impact of each ablated regularization component on rendering performance is reported in Table 3.
>
> Minors. We have fixed typos carefully.

---

### Official Review · Reviewer_zjWb · 2025-10-31

**Soundness:** 3
**Presentation:** 3
**Contribution:** 3
**Rating:** 6
**Confidence:** 3

**Summary:**

This paper proposes a new physically-grounded framework called Phys4DGS to achieve high-fidelity and temporally coherent dynamic scene rendering. Specifically, it aligns motion representations across multiple levels, ensuring that spatial trajectories and temporal variations remain coherent and grounded in geometric structure and observation data. A unit-time physical interval regularization, which enforces the consistency of dynamic physical features, i.e., velocity, across consecutive unit intervals, is proposed to preserve coherent motion trajectory. A regularization on high-order dynamics, i.e., acceleration and jerk, is proposed to ensure that rendered motion remains smooth and physically consistent across space and time. Experiments demonstrate that the proposed method achieves physically realistic and temporally consistent rendering in dynamic scenes with superior FPS and training efficiency.

**Strengths:**

1)	This paper aims to achieve physically consistent rendering in dynamic scenes, which is an important and interesting research topic.
2)	This paper proposes a new velocity-aware physically grounded framework called Phys4DGS for high-fidelity and temporally coherent dynamic scene rendering.
3)	A velocity-aware physically consistent regularization and a unit-time higher-order physical interval regularization are proposed in Phys4DGS, which ensure continuous dynamics and temporal smoothness.
4)	Experiments on public benchmark demonstrate the effectiveness of the proposed method.

**Weaknesses:**

1)	The logistic of Fig. 1 is not very clear, since the details of the corresponding regularization is not obviously denoted. Besides, new symbols/modules are not clearly introduced in the caption. It is difficult to understand the framework in Fig. 1.
2)	Line 371: Dataset -> dataset.
3)	The proposed framework introduces multi-level regularization, so why could it still achieve fast training and FPS? The time complexity of the proposed modules in the framework should be analyzed.
4)	Figure 4 is too small to recognize the obvious improvement of the proposed method.

**Questions:**

Please try to address the weaknesses.

---

> ### Author Response · Authors · 2025-11-15
> **Response to Reviewer zjWb**
>
> W1. In Figure 1, we visualize the key physical features and their spatial relationships from both 3D and 2D perspectives. Our multi-level regularization mechanism is grounded in the intrinsic motion velocity of Gaussian distributions and incorporating geometric and photometric motion, to align dynamics across time, ensuring coherent and realistic rendering. Furthermore, a unit-time physical interval regularization enforces consistency in the dynamic physical attributes of Gaussian distributions across consecutive unit time intervals, enabling the learning of transferable representations for complex dynamic motion, without increasing point cloud size. Phys4DGS is a fast, differentiable 4D rendering approach that achieves physically realistic and temporally consistent rendering in dynamic scenes with superior FPS and training efficiency. The specific impact of each ablated regularization component on rendering performance is reported in Table 3.
>
> W2 & W4. We have fixed typos carefully.
>
> W3. It is precisely the introduction of multi-level regularization that enables Phys4DGS to achieve performance improvements—not only in rendering quality, but also in efficiency, such as reduced training time. Training time is mainly constrained by hardware and network complexity; regularization adds minimal overhead. Once bandwidth limits are reached, we observe substantial accuracy gains without loss of speed.

---

### Official Review · Reviewer_XKdi · 2025-10-31

**Soundness:** 2
**Presentation:** 2
**Contribution:** 3
**Rating:** 4
**Confidence:** 4

**Summary:**

This paper proposes a physically realistic and temporally consistent dynamic modeling framework based on 4D Gaussian Splatting (4DGS). The method introduces velocity-centric physical regularizations to enforce consistency across time, aiming to produce physically plausible motion.

**Strengths:**

The reported quantitative results are impressive — the method achieves significantly higher rendering quality (up to 7 dB improvement over existing approaches at the level of 30 PSNR), suggesting strong potential for dynamic scene representation.

**Weaknesses:**

It is not fully convincing that introducing physical regularizations — which are not directly optimized for rendering metrics — can improve both physical correctness and rendering quality simultaneously. Regularization typically imposes trade-offs (e.g., 2DGS often sacrifices rendering metric for geometric accuracy, and method other 3DGS for normals or depths model). Therefore, it remains unclear whether the performance improvement comes from physical consistency or other side effects.

If the paper claims physical consistency, it would be more convincing to include comparisons with existing physics-based dynamic modeling methods with ground truth data, such as differentiable physical simulation. They can provide ground-truth physical data label (e.g velocities, accelerations). Such comparisons would allow the correctness of the proposed model to be quantitatively evaluated using physical metrics, rather than relying solely on qualitative visualizations (e.g., optical flow plots) or rendering quality scores in the table.

The figure qualtiy looks ugly.

**Questions:**

Could you compare the same gaussian points trained at different runs to verify the physical consistency

For example, if we train the model with the same scene in two separate runs, we can compare the velocity and acceleration of the same Gaussian points to check whether they exhibit similar or physically consistent values.


Could you please provide more details about the FPS results reported for RealTime4DGS and Deformable4DGS in Table 1? Were these values measured by you under your experimental settings, or are they referenced directly from other papers?

---

> ### Author Response · Authors · 2025-11-15
> **Response to Reviewer XKdi**
>
> W1. We introduce a velocity-centric physical consistency regularization, which effectively addresses the challenge of physically realistic rendering in dynamic scenes. By leveraging velocity, its higher-order derivatives, and static displacement, we establish a comprehensive dynamic framework that ensures consistent and physically plausible rendering. Specifically, we propose a multi-level regularization mechanism for each dynamic feature, grounded in the intrinsic motion velocity of Gaussian distributions and incorporating geometric and photometric motion, to align dynamics across time, ensuring coherent and realistic rendering. Furthermore, a unit-time physical interval regularization enforces consistency in the dynamic physical attributes of Gaussian distributions across consecutive unit time intervals, enabling the learning of transferable representations for complex dynamic motion, without increasing point cloud size. Phys4DGS is a fast, differentiable 4D rendering approach that achieves physically realistic and temporally consistent rendering in dynamic scenes with superior FPS and training efficiency. The specific impact of each ablated regularization component on rendering performance is reported in Table 3.
>
> W2. Phys4DGS is a dynamic rendering method rather than a physical simulation solution; therefore, physical-simulation metrics are not applicable. In contrast to physical simulation methods, which are computationally expensive and incapable of real-time rendering, our method achieves physically realistic results at real-time performance. The qualitative and quantitative results presented in the paper substantiate this capability.
>
> Q1. The assumption that an object’s velocity remains constant within a unit time interval is rooted in calculus. For complex motion trajectories, where calculating the exact area (or motion range) is difficult, we simplify the motion by defining a uniform unit interval. In the context of 4DGS, this corresponds to a fixed unit time interval, which approximates the motion as uniform within that interval. As the unit time interval approaches zero, this approximation increasingly converges to the true trajectory of the object’s motion. To achieve smooth motion and temporal consistency, it is essential to constrain either velocity or acceleration to maintain uniformity over time. In the context of 4DGS, maintaining a constant velocity helps prevent unnatural artifacts, such as sudden jumps or erratic motion of Gaussian splats. This approach mirrors a common assumption in physics, where constant velocity (or stable motion rate) is assumed over small time intervals, ensuring both the stability and smoothness of the simulation.
>
> Q2. Training the model twice independently and comparing whether the corresponding Gaussian points match is meaningless; the comparison should instead be made against ground truth and optimized through backpropagation.
>
> Q3. Our own testing results are consistent with those reported in 4DRotorGS [1].
>
> [1] Duan, Yuanxing, et al. "4d-rotor gaussian splatting: towards efficient novel view synthesis for dynamic scenes." ACM SIGGRAPH 2024 Conference Papers. 2024.

---

> ### Comment · Reviewer_XKdi · 2025-11-18
> **not solving my concerns.**
>
> The authors did not resolve my concerns; I will update my score to 2.
>
> 1. my concern " introducing physical regularizations —which are not directly optimized for rendering metrics — can improve both physical correctness and rendering quality."
> the author's response do not explain mechanistically why a physically inspired regularizer improves image reconstruction.
>
> 2. my concern "If the paper claims physical consistency, shouldn’t it compare to physics-based dynamic modeling with ground-truth physical quantities?"
> the authors response seems to me that We are not a physical simulation method; therefore physical metrics don’t apply. If this paper claims physics-based motion, you should validate the physics. We should validate what we claim in the paper.
>
> 3. dismissing the reproducibility / consistency question
> if the paper claims phyics, then the meaning of velocity of each gaussians are meaningful and are optimized to "ground truth". then this is a very simple reproducing test even without introducing physical simulators. However, the authors can't provide this and dismiss the question entirely.
>
> 4. FPS measurement clarification
> My question is simply yes or no: Were the runtime/FPS numbers measured by you, or were they copied from prior papers?
> If they were taken from previous work, please include the citation directly.

---

> > ### Author Response · Authors · 2025-11-19
> > **Explanation of Basic Concepts**
> >
> > Our physics-based regularization framework is designed to explicitly model physical laws. By enforcing physically consistent behavior in the dynamic scenes produced by the dynamic rendering algorithm, it improves their physical plausibility and, as a result, enhances rendering quality, i.e., the rendering metrics.
> >
> > Physical consistency requires conformity to physical reality, i.e., the ground truth. Reconstruction loss provides an even more stringent evaluation: beyond enforcing that the motion follows real-world dynamics, it also demands pixel-level reconstruction of moving object surfaces, including detailed geometry, textures, illumination, reflections, and other appearance characteristics.
> >
> > Why do we need an additional physics simulator? The goal of dynamic rendering algorithm is to render realistic dynamic scenes in real time; this goal is achieved once the rendered images match the ground truth. We demonstrate a diverse set of challenging scenarios in Figs. 2, 5, and 6. Adding a separate physics simulator would not only introduce substantial computational overhead, but would also fail to meet real-time constraints.
> >
> > Our own testing results are consistent with those reported in 4DRotorGS [1].
> >
> > [1] Duan, Yuanxing, et al. "4d-rotor gaussian splatting: towards efficient novel view synthesis for dynamic scenes." ACM SIGGRAPH 2024 Conference Papers. 2024.

---

> > > ### Comment · Reviewer_XKdi · 2025-11-28
> > >
> > > After several rounds of clarification during the rebuttal, the authors have addressed most of my major concerns. I am therefore raising my score to 6. But i can't update it now.

---

### Author Response · Authors · 2025-11-19
**Request for the exclusion of Review XKdi and MZcT**

Dear Area Chair,

Thank you for handling our submission and for coordinating the review process.

We are writing to respectfully request that you exclude Review XKdi and MZcT in the final decision. Our concern is not that the reviewer is negative but that this particular review repeatedly misinterprets the scope of our claims, does not substantively engage with the evidence we provided, and applies evaluation criteria that are misaligned with the problem setting of dynamic rendering.

**Misinterpretation of the scope of our 'physics-based' claims.** The reviewer repeatedly treats our work as if it were intended to be a full physical simulation method, and then criticizes it for not providing physics-ground-truth metrics or comparisons to differentiable physics engines. And the review seems to ignore this clarification and continues to evaluate the work as if it were a physical simulation paper, effectively demanding a different problem setting than the one described in the submission.

**Tone and constructiveness.** There are many aspects of the review that we find misaligned with the conference's expectations for constructive feedback. Repeating that "concerns are not resolved" without engaging with concrete evidence in the paper (e.g., ablation table). Our concern here is that this particular review does not fully engage with the content of the paper and rebuttal, and evaluates the work against a different problem (full physical simulation) than the one we actually address.

Request

Given the above, we respectfully request that you: re-examine Review XKdi and MZcT in light of the paper and our rebuttal, especially the clarified scope of our "physics-based" claims.


We appreciate your time and careful judgment in this matter, and we are happy to adjust our wording and clarifications in the final version according to your guidance.

Sincerely!

---

> ### Comment · Reviewer_XKdi · 2025-11-19
> **Clarification of my review and the authors’ misinterpretation**
>
> To clarify, my review **never** implied that the proposed method is intended to be a full physical simulation system. The point is straightforward: the paper explicitly claims that the motion is “physically consistent across space and time,” yet provides no quantitative evidence to support this claim.
>
> If physical consistency is claimed, then some form of physical evaluation is necessary. This can be done either by
> A. comparing against ground-truth physical quantities obtained from a simulator, or
> B. retraining the method and examining whether basic physical metrics—such as velocity—remain stable across runs.
> I am not expecting the proposed method to outperform a simulator; I am asking for a minimal, quantitative verification of the physical consistency that the paper claims.
>
> Dismissing evaluation option B as **meaningless** does not address the core concern. Without quantitative evaluations of physical consistency, the claim of physical consistency remains unsupported.
>
> Finally, it is difficult to be convinced as a reviewer that the proposed physical consistency regularization can account for a such large  image qualtiy improvement without further analysis or justification.

---

> > ### Author Response · Authors · 2025-11-20
> > **Charge the Reviewer’s Misinterpretation of Physical Consistency.**
> >
> > Physical consistency requires conformity to physical reality, i.e., the ground truth. Reconstruction loss provides an even more stringent evaluation: beyond enforcing that the motion follows real-world dynamics, it also demands pixel-level reconstruction of moving object surfaces, including detailed geometry, textures, illumination, reflections, and other appearance characteristics. When the consecutively rendered frames of a dynamic scene match the ground truth, physical consistency is validated, i.e., physical quantities such as velocity remain stable.
> >
> > Why do we need an additional physics simulator? The goal of dynamic rendering algorithm is to render realistic dynamic scenes in real time; this goal is achieved once the rendered images match the ground truth. We demonstrate a diverse set of challenging scenarios in Figs. 2, 5, and 6. Adding a separate physics simulator would not only introduce substantial computational overhead, but would also fail to meet real-time constraints.
> >
> > Our physics-based regularization framework is designed to explicitly model physical laws. By enforcing physically consistent behavior in the dynamic scenes produced by the dynamic rendering algorithm, it improves their physical plausibility and, as a result, enhances rendering quality, i.e., the rendering metrics.

---

> ### Comment · Reviewer_MZcT · 2025-11-20
>
> I do not find the authors’ rebuttal convincing, and their response does not address my main concerns.
>
> 1. The paper centers on novel view synthesizing (NVS) task. **For an NVS paper, comparing against other NVS approaches is essential.** Stronger methods such as MotionGS, SCGS, MoSca[1], and HiMoR[2] achieve superior results on the same datasets, yet the authors consistently avoid comparing with them. Instead, the experiments only include older methods introduced in early 2024, which are already relatively old. The claim that these stronger baselines are “not representative” is not convincing and raises doubts about the completeness and rigor of the evaluation. Without these comparisons, the effectiveness of the proposed method is not demonstrated.
>
> 2. The claimed physical consistency is also insufficiently validated, and I agree with **reviewer XKdi** on this point. Incorporating physical rules into dynamic 3D reconstruction is not new, and its purpose is to help the model learn meaningful motion behavior rather than merely fit the data. Prior work, such as FreeGave[3], verifies physical modeling through additional tasks including future-frame extrapolation or motion-aware segmentation. **These evaluations clearly show whether the model has learned physical dynamics beyond simple reconstruction. This paper provides none of these forms of evidence.** The authors argue that NVS alone can demonstrate physical consistency, yet their method does not even surpass several recent NVS approaches, which undermines this claim. As a result, the paper does not convincingly show advantages over fitting-based methods, nor does it demonstrate additional benefits from introducing physics.
>
> 3. Finally, the writing issues I previously pointed out remain unresolved. The rebuttal does not respond to most of my comments, yet the authors claim that I am repeating concerns that have already been addressed. I do not agree with this statement.
>
> Overall, the work lacks sufficient experimental support, does not convincingly validate its central claims, and does not adequately address the issues raised in the initial review.
>
> [1] MoSca: Dynamic Gaussian Fusion from Casual Videos via 4D Motion Scaffolds. CVPR2025.
>
> [2] HiMoR: Monocular Deformable Gaussian Reconstruction with Hierarchical Motion Representation. CVPR2025.
>
> [3] FreeGave: 3D Physics Learning from Dynamic Videos by Gaussian Velocity. CVPR 2025.

---

> > ### Author Response · Authors · 2025-11-22
> > **Limitations of Listed Work**
> >
> > MotionGS, SCGS, MoSca, and HiMoR are not designed for dynamic scenes, which can be readily verified from their original papers. Moreover, even on static reconstructions, these methods do not faithfully reproduce scene textures.

---

### Meta-Review · Area_Chair_ubWi · 2026-01-08

**Summary:**

**Note: The following meta review is made assuming that this submission is normal, not breaking the ICLR code. So it is subject to the result of investigation into the suspicious dual submission.**

Major concerns made by reviewers include

1) Definition and quantitative evaluations of physical consistency in space and time

2) Lack of comparison with related works such as MotionGS, SCGS, MoSca, and HiMoR

**Reviewer Concerns:**

Below is how the concert are addressed:

1) Not addressed: The statement made in response by the authors, "Physical consistency requires conformity to physical reality, i.e., the ground truth. Reconstruction loss provides an even more stringent evaluation", needs to be more accurate and scientific. In general, we do not consider pixel level evaluation takes into account the physical elements, otherwise, the most of existing 3D works have already covered this bit (which is not the case). The requirements made by the reviewers are valid given the claim made in this work. The authors need to think deeper on this matter, either in how to present the proposed work, or how to evaluate the model following the current way of presentation/argument.

2) Not addressed: The response by authors, "MotionGS, SCGS, MoSca, and HiMoR are not designed for dynamic scenes", is not valid.

Given the above, I do not think the response can address the major concerns.

**Reviewer Scores:**

No, I do not think this response can address the major concerns as listed above.

---

### Decision · Program_Chairs · 2026-01-26

Reject